# Efficient Multi-round LLM Inference over Disaggregated Serving

**Wenhao He** [1 2]  **Youhe Jiang** [3]  **Penghao Zhao** [4]  **Quanqing Xu** [5]  **Eiko Yoneki** [3]  **Bin Cui** [4 6]  **Fangcheng Fu** [1]

## Abstract

With the rapid evolution of Large Language Models (LLMs), multi-round workflows, such as autonomous agents and iterative retrieval, have become increasingly prevalent. However, this raises hurdles for serving LLMs under prefill-decode (PD) disaggregation, a widely adopted paradigm that separates the compute-bound prefill phase and memory-bound decode phase onto individual resources. Specifically, existing systems overlook the interleaved prefill-decode workload pattern in multi-round inference, leading to sub-optimal handling of the incremental prefill workloads and model deployment for the two phases.

In this work, we present AMPD, a brand new disaggregated serving framework for multi-round LLM inference. The core of AMPD is to coordinate the prefill workloads based on real-time workloads by adaptively determining *where* to carry out these workloads and *how* they are scheduled, in order to maximize service level objective (SLO) attainment. In addition, we tailor a planning algorithm for our scenario, facilitating the deduction of optimal resource allocation and parallel strategies for the two phases. Empirical results demonstrate that AMPD substantially improves SLO attainment compared to state-of-the-art baselines.

---

[1]School of Artificial Intelligence, Shanghai Jiao Tong University, Shanghai, China [2]Southeast University, Nanjing, China [3]Department of Computer Science, University of Cambridge, Cambridgeshire, UK [4]School of Computer Science & Beijing Key Laboratory of Software and Hardware Cooperative Artificial Intelligence Systems, Peking University [5]OceanBase, Ant Group, Hangzhou, China [6]Computational Social Science, Peking University (Qingdao), Qingdao, China. Correspondence to: Fangcheng Fu <ccchengff@sjtu.edu.cn>.

*Proceedings of the 43rd International Conference on Machine Learning*, Seoul, South Korea. PMLR 306, 2026. Copyright 2026 by the author(s).

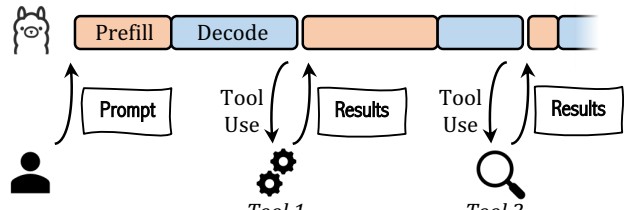

*Figure 1.* Illustration of multi-round LLM inference.

## 1. Introduction

Recent years have witnessed the remarkable success of Large Language Models (LLMs), demonstrating exceptional capabilities in natural language understanding and generation (Vaswani et al., 2017; Guo et al., 2025; Yang et al., 2025). Nowadays, LLMs have been widely deployed across diverse domains, ranging from code generation to creative writing. As these application scenarios continue to scale, how to deploy and serve LLMs efficiently has become critical for both academia and industry (Kwon et al., 2023; Zheng et al., 2024; NVIDIA, 2026b; Zhang et al., 2025c; Jiang et al., 2025a; Zhang et al., 2025b).

Generally, given a request, the LLM inference comprises two sequential phases: prefill and decode. The prefill phase processes the entire input prompt at once to compute the Key-Value (KV) cache, exhibiting compute-bound characteristics. In contrast, the decode phase generates tokens step-by-step auto-regressively, which is highly memory-bound. Given this distinct workload discrepancy, colocating both phases on the same hardware resources often leads to prefill-decode interference (a.k.a. resource contention) and low utilization. To address this, the prefill-decode (PD) disaggregation paradigm has emerged as a widely adopted solution (Zhong et al., 2024; Patel et al., 2024; Hu et al., 2024). By separating prefill and decode phases onto distinct hardware resources, PD disaggregation significantly improves LLM serving throughput and resource efficiency.

To accommodate more complex tasks, LLM inference has moved beyond single-shot responses to multi-round LLM workflows. Representative examples include autonomous agents (Yao et al., 2023) and iterative retrieval-augmented generation (RAG) (Yue et al., 2024), where the model continuously interacts with external environments (e.g., by functional calls). As shown in Figure 1, unlike single-shot in-

ference, these workflows require treating (part of, if not all) the output of environmental interactions as the input of subsequent generation. Consequently, from the perspective of LLM inference, these workflows necessitate an interleaved prefill-decode pattern, where incremental prefill computations occur between decoding steps throughout the inference process of a request.

Although research efforts have been made in both PD disaggregation and multi-round inference acceleration individually, the integration of them remains under-explored. Essentially, we identify that existing serving systems suffer from two critical limitations when applying PD disaggregation to multi-round workflows.

The first is the *lack of adaptive scheduling*. Specifically, in addition to the initial prefill and decode tasks, multi-round LLM inference further necessitates carrying out incremental prefill tasks (i.e., the prefill computations for subsequent rounds). To mitigate PD interference, all prefill[1] tasks are expected to be routed to the prefill instance(s). However, to meet service level objectives (SLOs), the serving system must dynamically decide two factors based on real-time workloads: (1) Whether an incremental prefill task should be executed locally on the decode instance, and if no, which prefill instance it should be routed to. (2) How to schedule the tasks on each prefill instance to cope with the increased burden brought by incremental prefill tasks. Existing systems lack adaptive mechanisms to handle these decisions to improve serving efficiency.

Secondly, there is a *lack of awareness for model deployment*. Specifically, the optimal configuration for model deployment (including resource allocation and parallel strategy) is strongly related to the workload characteristics. However, previous works determine the model deployment of each phase by merely considering the lengths of input prompts and output responses, which depict the prefill and decode workloads in single-round inference. Undoubtedly, this fails to take account of the unique interleaved workload pattern of multi-round inference, leading to sub-optimal resource provision and parallelism strategy deduction.

To fill this gap, this work presents AMPD, a novel serving framework for **A**daptive **M**ulti-round workflows with **PD** disaggregation by optimizing the runtime scheduling and deployment strategy. The technical contributions of this work are summarized as follows:

- We propose a runtime scheduling that addresses the unique interleaved workload pattern of multi-round inference. It consists of an adaptive routing strategy that dynamically decides where to execute prefill tasks based

on real-time system loads, as well as a prefill reordering policy that optimizes the execution order of queuing tasks to maximize SLO attainment.

- We develop an offline deployment planner that formulates the determination of resource allocation and parallel strategies as an integer linear programming problem, and solves it to deduce the optimal deployment configuration.

- We implement AMPD and evaluate it across diverse multi-round workloads. Extensive empirical results demonstrate that AMPD improves SLO attainment by 67.29%-339.74% on average compared to both co-located and disaggregated baselines.

## 2. Preliminaries and Related Works

**Prefill-decode disaggregation.** LLM inference consists of the compute-intensive prefill phase and the memory-bound decode phase. The time cost of first-token responsiveness, namely Time-to-First-Token (TTFT)[2], is dominated by the prefill phase, while the per-token generation time, namely Inter-Token Latency (ITL), is related to each decode step. In traditional LLM serving systems that co-locate both phases on the same GPU resources, the resource contention between the two phases leads to significant interference. Prefill-decode (PD) disaggregation (Zhong et al., 2024; Patel et al., 2024; Hu et al., 2024) is a promising technique to address such PD interference by dedicating GPU resources for each phase. Consequently, PD disaggregation has been widely adopted in LLM serving (Qin et al., 2025; Liu et al., 2024), and many efforts have been devoted to optimizing PD disaggregation from various aspects, including elastic scaling (Zhang et al., 2025a; Lai et al., 2025), mixed-GPU support (Jiang et al., 2025b), and multi-modal support (Singh et al., 2025; Dong et al., 2025). However, these works on PD disaggregation predominantly target single-round LLM inference, while our work has a different goal and can be incorporated with existing optimizations.

There are also approaches that jointly optimize aggregation and disaggregation (Ruan et al., 2025; Wang et al., 2025a). However, they do not consider the interleaved prefill-decode pattern in multi-round workflows. For instance, they assign part of the decode tasks to prefill-related instances, which is less suitable in our targeted scenario because prefill workloads are much heavier in multi-round workflows compared to single-round settings.

**Multi-round LLM workflows.** The capability of LLMs has evolved from simple, one-shot answering to complex, multi-round workflows. There are two representative scenarios. The first is autonomous agents, typically represented by

---

[1]Throughout this work, we explicitly distinguish "initial prefill" and "incremental prefill" when necessary. Unless specified otherwise, the term "prefill" refers to either or both variants.

[2]In this work, we use TTFT to measure the latency of both initial and incremental prefill.

the ReAct-style agentic workflows, which enable agents to interleave reasoning (LLM generation) with acting (tool use) (Yao et al., 2023; Mialon et al., 2023; Go & Park, 2025; Chen et al., 2024; Xu et al., 2026; Li et al., 2025a). The second is iterative retrieval-augmented generation (Shao et al., 2023; Yue et al., 2024; Jin et al., 2025), which actively retrieves new documents during generation to correct or refine the output. From the serving system's perspective, these multi-round workflows generate interleaved prefill-decode workloads.

**System optimizations for multi-round inference.** Recent works have proposed optimizations for multi-round LLM workflows. InferCept (Abhyankar et al., 2024) incorporates the discard, swap, and preserve operations when managing the history KV cache to reduce recomputation wastes. KVFlow (Pan et al., 2025) tailors the prefix caching for multi-agent workflows, allowing different agents to share common prefixes and thereby reduces redundant computation. vLLM-Continuum (Li et al., 2025b) predicts the duration of the environmental interactions and decides whether to retain the KV cache via a time-to-live mechanism. MARS (Shahout et al., 2025) and AugServe (Wang et al., 2025b) predict the output lengths and memory consumption respectively in multi-round LLM inference, and leverage such information to guide the scheduling of requests.

Although these works focus on how to optimize system performance for multi-round workflows, all of them are designed under the co-located serving paradigm. To achieve the holistic integration of multi-round workflows and PD disaggregation, there exist unexplored challenges of complex real-time task routing and scheduling as well as the disaggregated deployment configurations.

## 3. System Overview

Figure 2 illustrates the overview of AMPD, which comprises offline and online stages.

**Offline stage.** The offline stage profiles the workload characteristics and conducts the model deployment planning.

The *profiler* is responsible for measuring the execution time of different workloads, constructing a *performance model* for both our offline deployment planner and online adaptive scheduling. We adopt a piecewise $\alpha$-$\beta$ model to capture the performance characteristics of three types of tasks and form three time cost functions: (i) $\mathrm{T_{pre}}(l_{hist}, l_{incr}; \theta)$, which estimates the time cost of prefilling with history length $l_{hist}$ and incremental input length $l_{incr}$ under parallelism strategy $\theta$ (initial prefill tasks are associated with a history length of zero), (ii) $\mathrm{T_{dec}}(b; \theta)$, which estimates the time cost of decoding with batch size $b$ under parallelism strategy $\theta$, and (iii) $\mathrm{T_{kv}}(l_{ctx}; \theta_{src}, \theta_{dst})$, which estimates the time cost of transmitting KV cache with context length $l_{ctx}$ across the

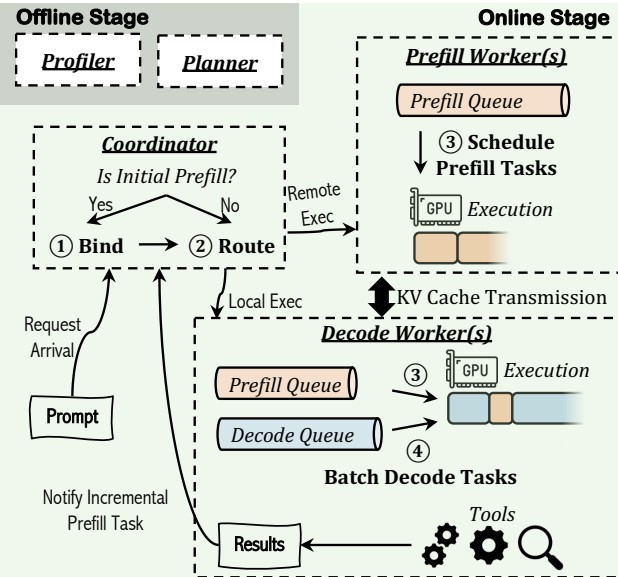

*Figure 2.* System overview of AMPD.

source and destination parallelism strategies ($\theta_{src} \rightarrow \theta_{dst}$).

The *planner* determines the optimal model deployment configuration (including the resource allocation and parallelism strategies for the two phases) prior to serving, by formulating and solving an *integer linear programming* problem.

**Online stage.** The online stage consists of two major components, namely the coordinator and workers.

The *coordinator* is in charge of the assignment of different tasks based on the real-time system loads.

The *prefill and decode workers* correspond to the two phases, respectively, maintaining task queues that record the metadata of pending tasks. Besides, each prefill/decode worker keeps recording its windowed TTFT/ITL statistic (by default, the average TTFT/ITL within the past 10 seconds). Furthermore, to facilitate up-to-date coordination, we implement these queues and windowed statistics with distributed shared memory so that they are globally accessible.

*Overall workflow.* During the online serving, the inference of a request undergoes the following steps.

① *Binding*: Upon the arrival of a request (a.k.a., a session), the coordinator first binds the request with a decode worker based on the memory usage (NVIDIA, 2026a). In other words, the decode worker will be responsible for managing the request's KV cache and conducting all its decode workloads. Such binding helps us unify the handling of both initial and incremental prefill tasks.

② *Routing*: Whenever the request requires prefilling (either its initial prefill after the binding or an incremental prefill after an environmental interaction), the coordinator employs an *adaptive routing mechanism* (§4.1) to determine where

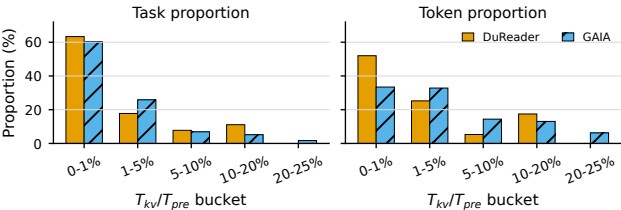

*Figure 3.* The estimated overhead of KV cache transfer divided by the estimated computation time of incremental prefill requests, summarized by the proportions by numbers of tasks and tokens.

to carry out this prefill task, with two kinds of choices.

- Local execution: The task is determined to be executed directly on the responsible decode worker (i.e., the decode worker that the request bound to). This avoids network transmission and reduces prefill workers' burden, but pauses the worker's ongoing decoding batch.[3]

- Remote execution: The task is routed to a prefill worker. This mitigates PD interference but necessitates KV transmission: (i) before the execution of this task, the prefill worker reads the history KV cache from the responsible decode worker if necessary; (ii) after the execution, the newly generated KV cache is transmitted back to the responsible decode worker.[4]

③ *Prefilling*: In either case, the designated worker enqueues the prefill task into its own prefill queue, and adopts a *prefill reordering policy* (§4.2) to schedule tasks in the prefill queue. If designated to a prefill worker, it performs KV transmission as described above.

④ *Decoding*: After step ③, the responsible decode worker possesses all KV cache of this request, and continues the auto-regressive decoding process.

In multi-round inference, the request may trigger an interaction with external environments (e.g., a function call), which is typically handled by a background CPU process of the decode worker. Upon the completion of interaction, it forms an incremental prefill task. Then, the coordinator is notified and we loop back to step ②. This recursion repeats until the inference of this request reaches termination.

## 4. Online Adaptive Scheduling

Our online adaptive request scheduling consists of two techniques, namely **adaptive routing** and **prefill reordering**.

---

[3]We follow the implementation in vLLM that prefill tasks are prioritized over decoding to avoid blocking subsequent generation.

[4]Each decode worker manages a local prefix cache, allowing it to merge the received incremental KV cache with the locally stored history KV cache. Thus, we only transmit the incremental KV cache to minimize network cost.

**Algorithm 1** Adaptive routing mechanism for prefill tasks. $\widehat{\mathrm{TTFT}}_i$ denotes the windowed TTFT statistic of the $i$-th prefill worker. $\widehat{\mathrm{ITL}}$ denotes the windowed ITL statistic of the current decode worker. $\mathrm{TTFT}_{thres}$ and $\mathrm{ITL}_{thres}$ are the thresholds for TTFT and ITL. $\alpha, \beta$ are hyper-parameters for identifying TTFT or ITL slack.

1: **for each** prefill worker $i$ in random order **do**
2:     **if** $\widehat{\mathrm{TTFT}}_i \leq \alpha \cdot \mathrm{TTFT}_{thres}$ **then**
3:         **return** remote$_i$
4: **if** $\widehat{\mathrm{ITL}} \leq \beta \cdot \mathrm{ITL}_{thres}$ **then**
5:     **return** local
6: Estimate local execution cost $t_{\mathsf{local}}$
7: **for each** prefill worker $i$ **do**
8:     Estimate remote execution cost $t_{\mathsf{remote}_i}$ via Eq. (2)
9: **return** $\arg\min_{route \in \{\mathsf{local}, \mathsf{remote}_1, \mathsf{remote}_2, \cdots\}} t_{route}$

These techniques address (i) where to carry out the prefill tasks, and (ii) how to schedule the prefill tasks within each worker, respectively.

### 4.1. Adaptive Routing

In essence, PD disaggregation separates the compute-intensive prefill phase and memory-bound decode phase onto individual resources, resolving the PD interference that occurs in co-located serving. Nevertheless, in multi-round inference, where to carry out the prefill tasks necessitates deliberation. For one thing, if we always route the prefill tasks (both initial and incremental) to prefill workers (remote execution), decode workers can remain focused on steady-state decoding and thus optimize ITL. However, it incurs extra transmission cost of KV cache, as shown in Figure 3. Moreover, it also increases the burden of prefill workers, harming TTFT. For another, carrying out some prefill workloads by the decode workers themselves (local execution) eliminates the transmission cost and benefit TTFT, but makes decode workers more susceptible to PD interference, which leads to spikes in ITL.

To strike a good balance between TTFT and ITL, we develop an *SLO-oriented* adaptive routing mechanism, which determines where each prefill task should be executed. The general idea is to leverage the windowed TTFT and ITL statistics to evaluate real-time system loads, and make the routing decision that is less likely to violate SLO requirements.

**Algorithm routine.** Algorithm 1 shows the workflow of our adaptive routing mechanism. Given a prefill task (along with the corresponding decode worker), we first identify prefill workers with sufficient slack to meet the TTFT SLO and route the task to an eligible candidate (lines 1-3). If all prefill workers are under pressure (i.e., TTFT approaching

the SLO threshold), we examine whether the current decoder worker's windowed ITL shows sufficient slack, and if yes, a local execution will be triggered (lines 4-5). Otherwise, we estimate and compare the time cost of both local and remote execution for the final routing decision (lines 6-9).

Particularly, as introduced in §3, we construct performance model based on offline profiling. Thus, the time cost of local execution on decode worker $d$ can be easily estimated via

$$t_{\text{local}} = \text{T}_{\text{pre}}(l_{hist}, l_{incr}; \theta_d) \quad + \\ \sum_{k \in \text{prefill\_queue}_d} \text{T}_{\text{pre}}(l_{hist_k}, l_{incr_k}; \theta_d), \quad (1)$$

where $l_{hist}$ denotes the history length, $l_{incr}$ denotes the incremental input length, $\theta_d$ denotes the parallelism strategy of decode worker $d$, and $\text{prefill\_queue}_d$ denotes its prefill queue. The first term is the estimated time cost of executing this prefill task on decode worker $d$ while the second term represents the estimated queuing time.

Similarly, if we route the task to the $i$-th prefill worker for remote execution, the time cost can be estimated via

$$t_{\text{remote}_i} = t_{\text{pre}_i} + t_{\text{kv}_i} + t_{\text{queue}_i}, \text{ where} \\ t_{\text{pre}_i} = \text{T}_{\text{pre}}(l_{hist}, l_{incr}; \theta_{p_i}), \\ t_{\text{kv}_i} = \text{T}_{\text{kv}}(l_{hist}; \theta_d, \theta_{p_i}) + \text{T}_{\text{kv}}(l_{incr}; \theta_{p_i}, \theta_d), \quad (2) \\ t_{\text{queue}_i} = \sum_{k \in \text{prefill\_queue}_i} \text{T}_{\text{pre}}(l_{hist_k}, l_{incr_k}; \theta_{p_i}),$$

where $\theta_{p_i}$ denotes the $i$-th prefill worker's parallelism strategy and $\text{prefill\_queue}_i$ denotes its prefill queue. The three terms correspond to the estimated time cost of (i) prefill computation, (ii) transmitting the KV cache back and forth, and (iii) queuing on the prefill worker.

Based on the estimated time cost, the prefill task will be routed to the one with the lowest cost.

**Discussion.** Although our adaptive routing needs to gather the shared metadata (windowed statistics and queuing information) from different workers, this is done concurrently. And the time cost estimation is extremely fast. Thus, the overhead of this mechanism is minor (detailed in §7).

### 4.2. Prefill Reordering

The routing mechanism in §4.1 leverages the TTFT/ITL slacks across prefill and decode workers to achieve real-time adaptive scheduling. In fact, thanks to our performance modeling, it is also feasible to identify the TTFT slacks of the tasks in each prefill queue. In other words, if a pending task is far from violating the TTFT SLO requirement, then we can schedule it later as a longer queuing time would not harm its TTFT SLO satisfaction.

Following this, we develop a lightweight *TTFT-aware* prefill reordering policy. In a nutshell, to schedule one task from

---

**Algorithm 2** Prefill reordering policy.

1: $W \leftarrow Q.\text{peek}(w), \pi^* \leftarrow \phi$
2: **for each** ordering $\pi$ over $W$ **do**
3:     **if** $\pi$ violates the postponement capacity **then**
4:        **continue**
5:     Compute the number of TTFT-satisfying tasks $S_\pi$ based on Eq (4).
6:     $\pi^* \leftarrow (\pi^* == \phi \text{ or } S_\pi > S_{\pi^*}) \text{ ? } \pi \text{ : } \pi^*$
7: Increment postponement counters for tasks that are postponed in $\pi^*$
8: $Q.\text{reorder}(\pi^*), r \leftarrow Q.\text{dequeue}()$
9: Schedule $r$ for execution

---

its prefill queue for execution, we employ a small lookahead window with size $w$ at the head of the queue, and reorder the tasks within the window in order to maximize the number of tasks satisfying the TTFT SLO.

**Algorithm routine.** The routine of our reordering policy is shown in Algorithm 2. Let $T_{now}$ denote the current time, $T_{\text{enq}}^{(r)}$ the enqueue time of task $r$, and $t_{\text{pre}}^{(r)}$ the estimated time cost for $r$ following our performance model. We first peek up to $w$ head elements of the prefill queue to form a window $W = \{r_1, r_2, \dots\}$ (line 1). Then, we enumerate feasible orderings to optimize TTFT within $W$ (lines 2-7). In particular, for any ordering $\pi$ over $W$, the completion time (relative to $T_{now}$) of the $\pi(k)$-th request (i.e., the $k$-th one in $\pi$) can be predicted as

$$C^{(\pi(k))} = \sum_{j=1}^{k} t_{\text{pre}}^{(\pi(j))}. \quad (3)$$

Subsequently, the number of TTFT-satisfying tasks within $W$ under ordering $\pi$ can be predicted as

$$S_\pi = \sum_{k=1}^{|W|} \mathbb{I}\Big[ (T_{now} - T_{\text{enq}}^{(\pi(k))}) + C^{(\pi(k))} \leq \text{TTFT}_{\text{thres}} \Big]. \quad (4)$$

Consequently, to improve TTFT SLO attainment, we choose the ordering $\pi^* = \arg\max_\pi S_\pi$ that maximizes the number of TTFT-satisfying tasks within $W$ (lines 5-6).

In addition, we enforce that each task can be postponed by at most $w$ times to prevent starvation. Particularly, we maintain a *postponement counter* for each task, and increment it whenever the task is postponed due to reordering (line 7). Once there are any tasks' postponement counters reaching $w$, we skip the orderings that postpone such tasks (lines 3-4).

After the optimal ordering is obtained, the head elements of the prefill queue are reordered accordingly, after which, the first element will be scheduled for execution (lines 8-9).

**Discussion.** Since it is extremely fast to evaluate the number

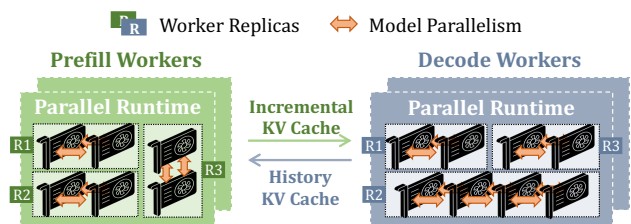

*Figure 4.* Illustration of disaggregated LLM serving where prefill and decode workers are with different parallelism configurations.

of TTFT-satisfying tasks for one ordering and the window size is typically small (less than 5 in practice), the overhead incurred by reordering is negligible (detailed in §7).

Last but not least, it is noteworthy that the estimated queuing time in Eq. (2) does not take account of the prefill reordering policy. Nevertheless, it does not harm the effectiveness of our adaptive routing mechanism, since the window size is small compared to the size of the prefill queue in practice.

## 5. Offline Deployment Planning

As depicted in Figure 4, the deployment of disaggregated serving should consider two factors for each worker type (prefill and decode): (i) instantiating multiple independent replicas to handle concurrent requests (a.k.a., data parallelism (Li et al., 2023)), and (ii) partitioning each individual replica across multiple GPUs to satisfy memory and computational requirements (a.k.a., model parallelism (Shoeybi et al., 2019)). Consequently, to optimize the model deployment, we must establish appropriate data and model parallelism configurations for both worker types, while conforming to a fixed GPU resource capacity constraint.

In response to this, we formulate the model deployment planning as a *resource-constrained optimization problem*. The goal is to identify the specific combination of model parallelism (partition) and data parallelism (replication) configurations for each worker type that maximizes performance.

**Formulation of planning.** We first introduce the *decision variables*. Since model parallelism degrees are typically powers of 2, we can represent the deployment via the number of replicas associated with each feasible model parallelism degree. Specifically, given a discrete set of supported model parallelism degrees $\mathcal{T}$ (e.g., $\{1, 2, 4, 8\}$), the decision variables comprise two integer vectors, $\mathbf{x}, \mathbf{y} \in \mathbb{Z}_{\geq 0}^{|\mathcal{T}|}$ (e.g., $\mathbf{x} = [x^{(1)}, x^{(2)}, x^{(4)}, x^{(8)}]$, $\mathbf{y} = [y^{(1)}, y^{(2)}, y^{(4)}, y^{(8)}]$). For each model parallelism degree $n \in \mathcal{T}$, the variable $x^{(n)}$ ($y^{(n)}$) denotes the number of prefill (decode) workers that are deployed with model parallelism degree $n$ (i.e., each partitioned across $n$ GPUs).[5]

---

[5]Our formulation considers heterogeneous TP sizes to allow for flexible, irregular GPU allocation across phases.

Subsequently, to establish the optimization target, we introduce an auxiliary variable $Z$, which represents the maximum P95 latency among all workers. We formulate this resource allocation problem as an Integer Linear Programming (ILP) problem (Vielma, 2015):

$$\underset{\mathbf{x}, \mathbf{y} \in \mathbb{Z}_{\geq 0}^{|\mathcal{T}|}}{\arg\min} \quad Z$$

s.t. **(C1)** $Z \geq \tau_{pre}(n), \quad \forall n \in \mathcal{T}$ where $x^{(n)} \geq 1$

**(C2)** $Z \geq \tau_{dec}(n), \quad \forall n \in \mathcal{T}$ where $y^{(n)} \geq 1$   (5)

**(C3)** $\sum_{n \in \mathcal{T}} (x^{(n)} \cdot n) + \sum_{n \in \mathcal{T}} (y^{(n)} \cdot n) \leq N$

Constraints **(C1)** and **(C2)** establish P95 latency bounds of disaggregated inference. The coefficients $\tau_{\text{pre}}(n), \tau_{\text{dec}}(n)$ represent the estimated P95 latency[6] of a single prefill or decode worker replica configured with model parallelism degree $n$. Since the two constraints require that $Z$ must be at least as large as the P95 latency of any instantiated worker replica, minimizing $Z$ represents minimizing the worst-case P95 latency across all deployed worker replicas (i.e., optimizing the bottlenecked worker replicas). Constraint **(C3)** enforces the global GPU resource capacity. For each configuration with model parallelism degree $n$, the GPU consumption equals the product of the number of worker replicas ($x^{(n)}$ or $y^{(n)}$) and the parallelism degree $n$. This constraint ensures that the total number of GPUs allocated across all worker replicas does not exceed the available cluster capacity $N$.

**Planning solver.** The formulation in Eq. (5) constitutes a multi-dimensional variant of the *Unbounded Knapsack Problem* (Pisinger & Toth, 1998), requiring the solver to evaluate integer combinations of valid model parallelism degrees to identify the optimal resource allocation. We employ a standard Mixed Integer Linear Programming (MILP) solver (Huangfu & Hall, 2018) to solve this optimization problem. The solver accepts the simulated P95 latency coefficients $\tau_{\text{pre}}(n)$ and $\tau_{\text{dec}}(n)$ as constant parameters, and explores the feasible region defined by the integer vectors $\mathbf{x}$ and $\mathbf{y}$. This approach is computationally efficient for typical cluster scales (see Appendix A.2 for details), and guarantees identification of the global optimum that balances both worker types while fully utilizing available GPU resources.

**Discussion.** In Eq. (5), we adopt the P95 latency as our surrogate objective, while the goal of our system (particularly, the two approaches in §4) is to maximize SLO attainment. This discrepancy is due to the infeasibility to solve the optimization of SLO attainment via efficient linear programming solvers. Particularly, owing to the binary output of SLO satisfaction, the SLO attainment metric cannot

---

[6]We adopt a simulator to estimate the P95 latency of different configurations based on the performance model as well as the techniques proposed in §4, which is detailed in Appendix A.1.

*Table 1.* The average number of rounds, prefill length, and decode length of each trace.

| Trace | #Rounds | Prefill Length | Decode Length |
|---|---|---|---|
| ToolBench | 3.96 | 703.79 | 50.39 |
| GAIA | 11.32 | 6161.02 | 528.76 |
| HotpotQA | 3 | 1569.8 | 80.03 |
| DuReader | 3 | 3081.23 | 150.10 |

serve as a continuous surrogate objective. Consequently, our ILP is formulated to minimize the P95 latency. Despite this discrepancy, empirical evaluation demonstrates that our planning is effective in finding the optimal deployment. We refer interested readers to Appendix A.3 for more details.

# 6. Implementation

AMPD is implemented atop NVIDIA Dynamo (NVIDIA, 2026b). We adopt Redis to enable global sharing of the queues and windowed TTFT/ITL statistics, and utilize the SCIP library (Bolusani et al., 2024) to solve Eq. (5). For efficient KV cache transmission, we leverage the NVIDIA Inference Xfer Library (NIXL) (NVIDIA, 2026c) to achieve point-to-point RDMA network communication across prefill and decode workers. The transmission of history KV cache is implemented as lazy reads. To be specific, our routing mechanism (§4.1) sends merely metadata to prefill workers for remote execution. Only when an incremental prefill task is scheduled for execution (§4.2), should the prefill worker read the history KV cache from the decode worker. Additionally, the KV cache transmission overlaps with the computation of the previous task to hide the network latency.

# 7. Experiments

## 7.1. Experimental Setup

**Environments.** We conduct experiments over 4 servers, each with 8 NVIDIA H20 (96GB) GPUs. The GPUs within each server are interconnected via NVLink with a bandwidth of 900GB/s, while the servers are interconnected by InfiniBand with a bandwidth of 200GB/s.

**Workloads.** Three LLMs are used in our experiments, which are Qwen3-32B (Yang et al., 2025), Llama3.1-70B (Grattafiori et al., 2024), and Mixtral-8x7B (Jiang et al., 2024a). Since our major focus is serving efficiency, we conduct experiments by running the same traces with AMPD and the baselines to achieve a fair comparison. Four workload traces are considered, which are generated from the ToolBench (Guo et al., 2024), GAIA (Mialon et al., 2024), HotpotQA (Yang et al., 2018), and DuReader (He et al., 2018) datasets, respectively. Due to the space constraint, we summarize the length information in Table 1, while leaving the details of these traces to Appendix B. For each model,

we run HotpotQA and ToolBench workloads on a single server (8 GPUs), DuReader on two servers (16 GPUs), and GAIA on four servers (32 GPUs), to match the workload scale and system capacity.

**Baselines.** We compare AMPD with three state-of-the-art LLM serving systems as follows.

- Dynamo (NVIDIA, 2026b): The original NVIDIA Dynamo disaggregated serving system under PD disaggregation without the techniques proposed in this work. It always separates the prefill and decode workloads onto different resources.

- vLLM (Kwon et al., 2023): One of the most prestigious LLM serving system under PD co-location.

- vLLM-Continuum (Li et al., 2025b): A serving system based on vLLM tailored for multi-round workflows.

**Protocols.** In all experiments, we utilize our offline planner to determine the model deployment of AMPD, while we tune the model deployment for all baselines and report the best results to be fair. By default, the hyper-parameters are set as $\alpha = 0.9, \beta = 0.85, w = 3$. We follow the previous standard to generate request arrival times using Poisson process with different request arrival rates (Kwon et al., 2023; Zhong et al., 2024; Yu et al., 2022), and take SLO attainment as the primary evaluation metric.

## 7.2. Experiment Results

**End-to-end comparison.** We conduct experiments to measure the SLO attainment of all counterparts under various request arrival rates, with results presented in Figure 5.

Overall, AMPD achieves the highest SLO attainment across all experiments. Compared to Dynamo (the disaggregated baseline) and vLLM (the co-located baseline), AMPD improves SLO attainment by up to 967.54% and 3435.1% (67.29% and 339.74% on average), respectively.

To understand the performance gain, we further provide a detailed breakdown in Figure 6. It can be observed that both co-located and disaggregated serving exhibit divergent latency trade-offs. Co-located serving favors executing prefill tasks earlier (e.g., via prioritization or preemption) for the sake of continuous batching, which reduces TTFT but disrupts the execution of decode tasks and thereby deteriorates ITL. In contrast, PD disaggregation decouples prefill from decode, reducing PD interference on decode workers and typically achieving lower ITL, yet TTFT often becomes the bottleneck due to the reduction in prefill resources and the overhead of KV cache transmission. These trends are also observed across tasks. On HotpotQA, DuReader, and Tool-Bench, disaggregated serving often outperforms co-located serving because its ITL advantage dominates end-to-end

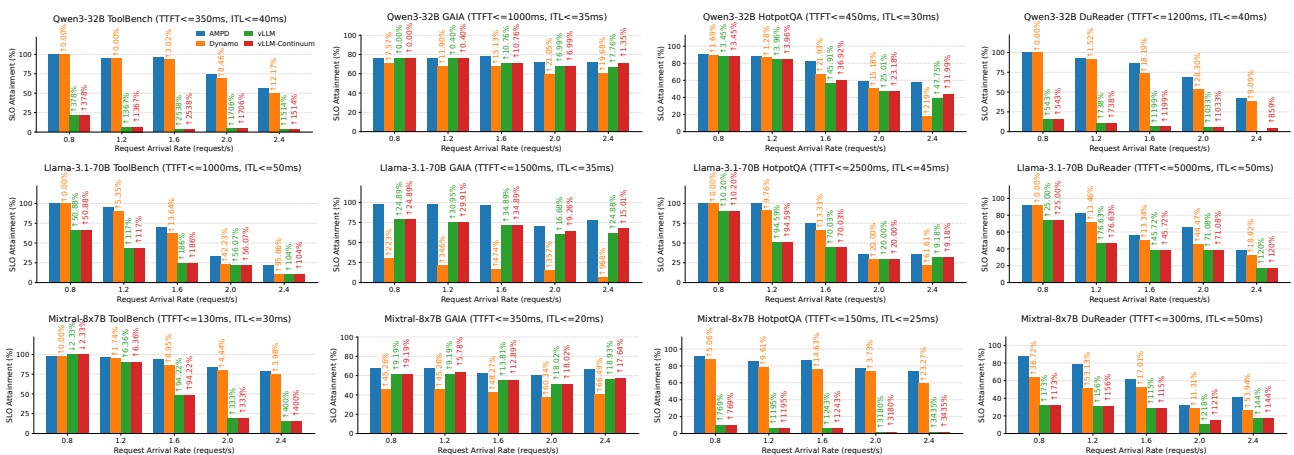

*Figure 5.* End-to-end comparison. Each row reports the SLO attainment under different traces and request arrival rates for one model.

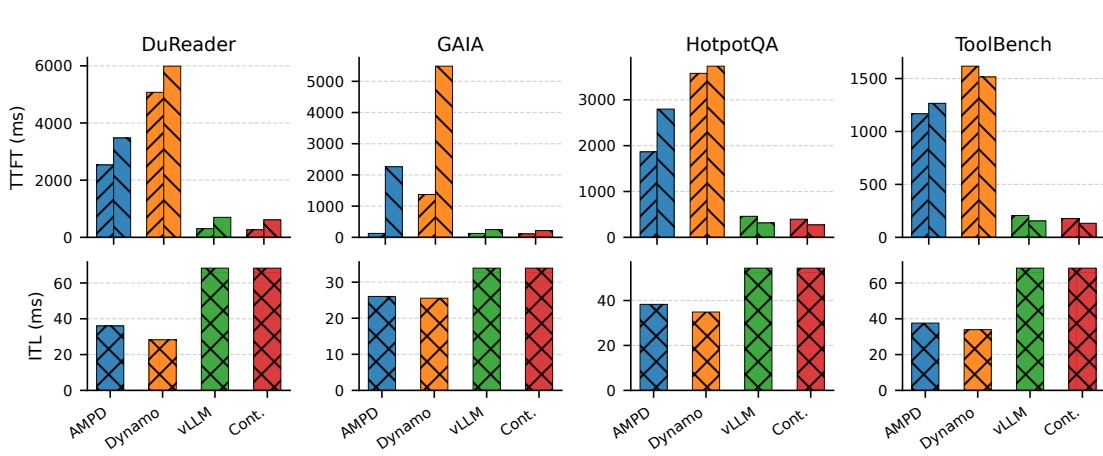

*Figure 6.* Detailed latency breakdown of Llama3.1-70B at 2 reqs/s, including the average TTFT for initial prefill, average TTFT for incremental prefill, and average ITL for decoding.

behavior. On GAIA, co-located serving becomes favorable (compared to Dynamo) since it provides competitive TTFT while not exhibiting a pronounced ITL disadvantage. Consequently, both vLLM and Dynamo are frequently dominated by either ITL or TTFT constraints, limiting the attainable SLO rate.

AMPD enjoys the benefit of PD disaggregation, avoiding deteriorating ITL compared to vLLM. Meanwhile, compared to Dynamo, AMPD significantly reduces TTFT thanks to our adaptive scheduling. Consequently, AMPD achieves a more robust trade-off between first-token responsiveness and per-token generation speed.

vLLM-Continuum follows the co-located architecture and adopts a queuing policy that prioritizes tasks within the same request (a.k.a., session). Since such tasks can reuse cached KV states, they typically require less computation, which shortens queuing delay and reduces TTFT while leaving ITL unaffected. However, in many settings, the TTFT reduction does not consistently translate into a substantial improvement in SLO attainment. Thus, vLLM-Continuum

is often comparable to, or nearly tied with, vLLM.

**Overhead of AMPD.** We further measure the overhead of the techniques proposed in §4. Across the experiments, the total time cost of adaptive routing is 0.39 ms on average, with metadata reading (via distributed shared memory) consuming merely 0.15 ms, and the total time cost of prefill reordering is 0.45 ms on average. Since LLM prefill computations and decoding steps typically take tens to hundreds of milliseconds, the end-to-end overhead of adaptive routing and prefill reordering is negligible. Moreover, they run on CPU and overlap with GPU computation.

**Ablation study.** We conduct experiments to assess the effectiveness of the two techniques in our online scheduling. The results are provided in Figure 7. The adaptive routing mechanism routes 13.9%-31.7% prefill tasks to be executed locally on the decode workers, which relieves the burden of prefill workers, and thereby increases the SLO attainment by 27.37%-350%, demonstrating the effectiveness of adaptive routing in mitigating workload imbalance between phases. In addition, the prefill reordering policy further improves

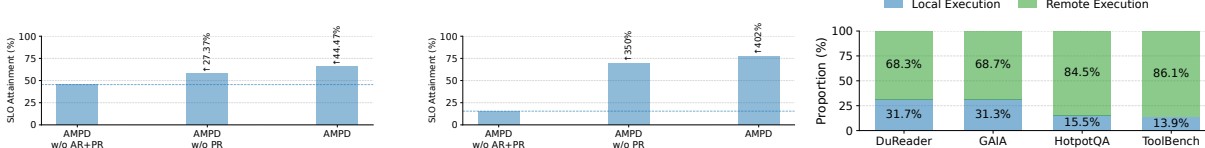

*Figure 7.* Ablation studies (Llama3.1-70B, 2 reqs/s). Left: DuReader. Middle: GAIA. Right: Proportion of local and remote execution.

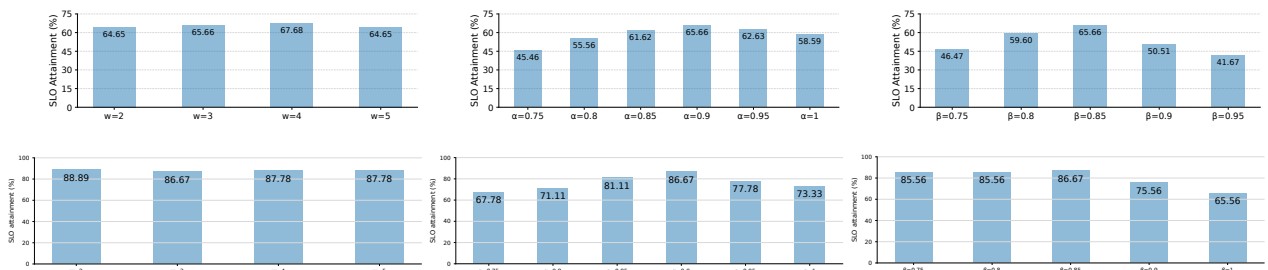

*Figure 8.* Sensitivity experiments w.r.t. $w, \alpha, \beta$. (Top: Llama3.1-70B, DuReader, 2 reqs/s. Bottom: Qwen3-32B, HotpotQA, 1.2 req/s.)

the SLO attainment by 13.42%-14.81%. This is reasonable as the queuing optimization helps reduce the tail of TTFT. Eventually, these two techniques together provision 44.47%-402% of improvement.

**Sensitivity.** Both the two techniques in online scheduling introduce additional hyper-parameters. We evaluate the sensitivity of our work by varying these hyper-parameters. The results are shown in Figure 8.

We first evaluate the SLO attainment with multiple window sizes (2, 3, 4, and 5) in prefill reordering. It can be seen that the performance of AMPD remains similar across these window sizes (within 3% gap). This suggests that a short time window is sufficient to capture load signals needed for effective routing decisions.

Next, we evaluate the impact of $\alpha$ and $\beta$, which control the sensitivity to congestion on the prefill and decode sides, respectively. In general, a smaller $\alpha$ enables earlier detection of prefill congestion but may lead to under-utilization of prefill workers, while a smaller $\beta$ triggers earlier detection of decode congestion at the cost of potentially reduced decode utilization. Consequently, moderate values of $\alpha$ and $\beta$ can strike a better balance between prefill and decode workers, improving the overall system efficiency.

## 8. Conclusion and Possible Future Works

We presented AMPD, a novel disaggregated serving framework for efficient multi-round LLM inference. AMPD addresses the interleaved prefill-decode workload patterns of multi-round inference by runtime adaptive scheduling, consisting of the adaptive routing mechanism and prefill reordering policy to balance prefill and decode loads. Extensive experiments show that, compared to existing works, AMPD improves SLO attainment by 67.29%-339.74% on

average, provisioning an effective solution for serving complex, multi-round LLM workflows.

Lastly, we would like to discuss possible directions for strengthening our work. The first is to integrate AMPD with the chunked prefill technique (Agrawal et al., 2024). Specifically, by breaking long prefill requests into smaller chunks during local execution, we can reduce decode blocking and thereby improve SLO attainment. Secondly, our offline planner currently takes as input a fixed workload and hardware condition, so the deduced deployment may become sub-optimal when workload drifts or hardware availability changes (e.g., elastic scaling or fault recovery). To solve this, we can incorporate the monitor-and-replan approach that used in prior works (Zhong et al., 2024). Moreover, we can run the replanning process on a background CPU thread and perform rolling migration (adjusting replicas one by one) to avoid halting the online service.

## Acknowledgments

This work is supported by National Natural Science Foundation of China (62402011, U23B2048), Fundamental and Interdisciplinary Disciplines Breakthrough Plan of the Ministry of Education of China (JYB2025XDXM108), and the Electronics Society-OceanBase Database Research Special Project. Fangcheng Fu is the corresponding author.

## Impact Statement

This paper presents work whose goal is to advance the field of Machine Learning. There are many potential societal consequences of our work, none which we feel must be specifically highlighted here.

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

# A. More Details about Offline Planning

## A.1. Performance Simulation

The simulator generates simulated performance metrics (i.e., P95 latency) for various model deployment configurations.

**Simulator inputs.** In addition to the target model deployment configuration, the simulator requires three categories of inputs: (**i**) *Model specifications*, including the architecture type (e.g., dense or sparse) and parameter configurations (e.g., number of layers and hidden size); (**ii**) *Hardware characteristics*, encompassing GPU specifications such as memory capacity and inter-GPU communication bandwidth; and (**iii**) *Workload characteristics*, including request arrival rates and input/output sequence lengths. Collectively, these inputs enable the simulator to accurately model the serving behavior under specific deployment scenarios.

**Simulation process.** Given the aforementioned inputs, the simulation proceeds in two stages: The *profiling stage* and the *execution stage*. (i) The profiling stage is in the charge of the profiler mentioned in §3. It enumerates all operators within the model (including the Q, K, V linear projections, self-attention computations, and feed-forward layers) and profiles the execution time of each operator on the target hardware. (ii) During the execution stage, the simulator leverages the operator latencies obtained from the profiling stage to simulate the concurrent execution of multiple incoming requests. Key inference serving features, such as request dispatching (Katevenis et al., 1991), continuous batching (Yu et al., 2022), model replication (Li et al., 2023), and model parallelism (Shoeybi et al., 2019), are incorporated into this simulation process. The simulator ultimately outputs the P95 latency for the target model deployment configuration.

**Simulation for phase splitting and KV retransmission.** To accurately evaluate deployment scenarios involving prefill-decoding disaggregation and KV retransmission, we explicitly integrate these features into our simulator. (**i**) *Phase splitting*: The simulator supports deploying prefill and decoding replicas onto distinct GPU resources, enabling independent configurations of model parallelism and GPU allocation for each phase. The KV transmission latency between phases is modeled using the $\alpha$-$\beta$ model (Hockney, 1994). (**ii**) *KV retransmission*: The simulator incorporates a decision-making mechanism to dynamically determine whether to offload incremental prefill to the prefill model replica or execute it locally on the decoding model replica. This mechanism implements the scheduling policy presented in §4.

## A.2. Time Cost of Planning

Figure 9 measures the time cost of our offline planning with different number of GPUs. Since the determination of deployment configuration is formulated as an Integer Linear Programming problem, it can be solved efficiently with existing solvers (Bolusani et al., 2024). Consequently, our planning finishes quickly, taking merely one minute over 256 GPUs. This represents a common cluster scale for LLM serving in many downstream applications.

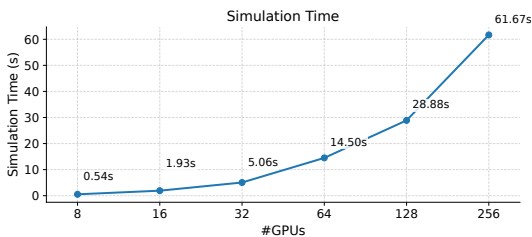

*Figure 9.* Time cost of offline planning with varying numbers of GPUs.

## A.3. Effectiveness of Planning

Table 2 shows the top-3 ranking of deployment configurations based on our planner's deduction and actual serving. It shows that our planner successfully finds out the optimal configuration that empirically achieves the best performance.

As discussed in §5, there is a discrepancy between the objective of our formulation (P95 latency) and system performance (SLO attainment). Intuitively, when the TTFT and ITL thresholds are extremely strict or extremely loose, such a discrepancy may cause the offline planner's solution to deviate from the actual optimum. In these cases, the binary nature of the SLO metric dominates: the vast majority of deployment strategies will yield an SLO attainment of nearly 0% or 100%. Thus, optimizing the continuous P95 latency metric will fail to translate into meaningful improvement in SLO attainment. However, in practical LLM serving scenarios, TTFT and ITL thresholds are usually set to reasonable targets, so our method is empirically reliable, as evidenced by the results in Table 2.

*Table 2.* Top-3 model deployment configurations deduced by our planner and real-system serving. Across all experiments, our planner produces identical configurations to real-time serving.

| Model | Trace | Rank | Planner | Real-system Serving |
|---|---|---|---|---|
| Llama3.1-70B | DuReader | [#1] | P:<TP=4, DP=2>, D:<TP=8, DP=1> | P:<TP=4, DP=2>, D:<TP=8, DP=1> |
| | | [#2] | P:<TP=2, DP=4>, D:<TP=8, DP=1> | P:<TP=2, DP=4>, D:<TP=8, DP=1> |
| | | [#3] | P:<TP=8, DP=1>, D:<TP=8, DP=1> | P:<TP=8, DP=1>, D:<TP=8, DP=1> |
| | HotpotQA | [#1] | P:<TP=4, DP=1>, D:<TP=4, DP=1> | P:<TP=4, DP=1>, D:<TP=4, DP=1> |
| | | [#2] | P:<TP=2, DP=2>, D:<TP=4, DP=1> | P:<TP=2, DP=2>, D:<TP=4, DP=1> |
| | | [#3] | P:<TP=2, DP=1>, D:<TP=2, DP=3> | P:<TP=2, DP=1>, D:<TP=2, DP=3> |
| | GAIA | [#1] | P:<TP=8, DP=2>, D:<TP=8, DP=2> | P:<TP=8, DP=2>, D:<TP=8, DP=2> |
| | | [#2] | P:<TP=8, DP=3>, D:<TP=8, DP=1> | P:<TP=8, DP=3>, D:<TP=8, DP=1> |
| | | [#3] | P:<TP=4, DP=4>, D:<TP=8, DP=2> | P:<TP=4, DP=4>, D:<TP=8, DP=2> |
| | ToolBench | [#1] | P:<TP=4, DP=1>, D:<TP=4, DP=1> | P:<TP=4, DP=1>, D:<TP=4, DP=1> |
| | | [#2] | P:<TP=2, DP=2>, D:<TP=4, DP=1> | P:<TP=2, DP=2>, D:<TP=4, DP=1> |
| | | [#3] | P:<TP=2, DP=1>, D:<TP=2, DP=3> | P:<TP=2, DP=1>, D:<TP=2, DP=3> |
| Qwen3-32B | HotpotQA | [#1] | P:<TP=4, DP=1>, D:<TP=4, DP=1> | P:<TP=4, DP=1>, D:<TP=4, DP=1> |
| | | [#2] | P:<TP=2, DP=2>, D:<TP=4, DP=1> | P:<TP=2, DP=2>, D:<TP=4, DP=1> |
| | | [#3] | P:<TP=2, DP=1>, D:<TP=2, DP=3> | P:<TP=2, DP=1>, D:<TP=2, DP=3> |
| | DuReader | [#1] | P:<TP=4, DP=2>, D:<TP=4, DP=2> | P:<TP=4, DP=2>, D:<TP=4, DP=2> |
| | | [#2] | P:<TP=4, DP=2>, D:<TP=2, DP=4> | P:<TP=4, DP=2>, D:<TP=2, DP=4> |
| | | [#3] | P:<TP=2, DP=4>, D:<TP=4, DP=2> | P:<TP=2, DP=4>, D:<TP=4, DP=2> |
| | ToolBench | [#1] | P:<TP=4, DP=1>, D:<TP=4, DP=1> | P:<TP=4, DP=1>, D:<TP=4, DP=1> |
| | | [#2] | P:<TP=2, DP=2>, D:<TP=4, DP=1> | P:<TP=2, DP=2>, D:<TP=4, DP=1> |
| | | [#3] | P:<TP=2, DP=1>, D:<TP=2, DP=3> | P:<TP=2, DP=1>, D:<TP=2, DP=3> |
| | GAIA | [#1] | P:<TP=8, DP=2>, D:<TP=8, DP=2> | P:<TP=8, DP=2>, D:<TP=8, DP=2> |
| | | [#2] | P:<TP=8, DP=3>, D:<TP=8, DP=1> | P:<TP=8, DP=3>, D:<TP=8, DP=1> |
| | | [#3] | P:<TP=4, DP=4>, D:<TP=8, DP=2> | P:<TP=4, DP=4>, D:<TP=8, DP=2> |
| Mixtral-8x7B | HotpotQA | [#1] | P:<TP=4, DP=1>, D:<TP=4, DP=1> | P:<TP=4, DP=1>, D:<TP=4, DP=1> |
| | | [#2] | P:<TP=2, DP=1>, D:<TP=2, DP=3> | P:<TP=2, DP=1>, D:<TP=2, DP=3> |
| | | [#3] | P:<TP=4, DP=1>, D:<TP=2, DP=2> | P:<TP=4, DP=1>, D:<TP=2, DP=2> |
| | DuReader | [#1] | P:<TP=8, DP=1>, D:<TP=8, DP=1> | P:<TP=8, DP=1>, D:<TP=8, DP=1> |
| | | [#2] | P:<TP=8, DP=1>, D:<TP=4, DP=2> | P:<TP=8, DP=1>, D:<TP=4, DP=2> |
| | | [#3] | P:<TP=4, DP=2>, D:<TP=8, DP=1> | P:<TP=4, DP=2>, D:<TP=8, DP=1> |
| | ToolBench | [#1] | P:<TP=4, DP=1>, D:<TP=4, DP=1> | P:<TP=4, DP=1>, D:<TP=4, DP=1> |
| | | [#2] | P:<TP=2, DP=1>, D:<TP=2, DP=3> | P:<TP=2, DP=1>, D:<TP=2, DP=3> |
| | | [#3] | P:<TP=4, DP=1>, D:<TP=2, DP=2> | P:<TP=4, DP=1>, D:<TP=2, DP=2> |
| | GAIA | [#1] | P:<TP=8, DP=2>, D:<TP=8, DP=2> | P:<TP=8, DP=2>, D:<TP=8, DP=2> |
| | | [#2] | P:<TP=8, DP=3>, D:<TP=8, DP=1> | P:<TP=8, DP=3>, D:<TP=8, DP=1> |
| | | [#3] | P:<TP=4, DP=4>, D:<TP=8, DP=2> | P:<TP=4, DP=4>, D:<TP=8, DP=2> |

## B. More Details of Experiments

**Experimental traces.** The traces used in our experiments are generated from public datasets. In particular, for ToolBench[7] and GAIA[8], we adopt publicly available traces to represent agentic workflows, while for HotpotQA and DuReader, we record one trace for each dataset by running iterative retrieval augmented generation (Yue et al., 2024) using Qwen3-32B, with each request invoking three retrieval calls.

**More experimental results.** Figure 10 compares the average end-to-end latency of all counterparts. AMPD maintains low latencies that are comparable against Dynamo. Although Dynamo has lower latencies in some cases, the gap is small. More importantly, AMPD delivers substantial improvement in terms of SLO attainment (as evaluated in §7), which is essential for real-world serving.

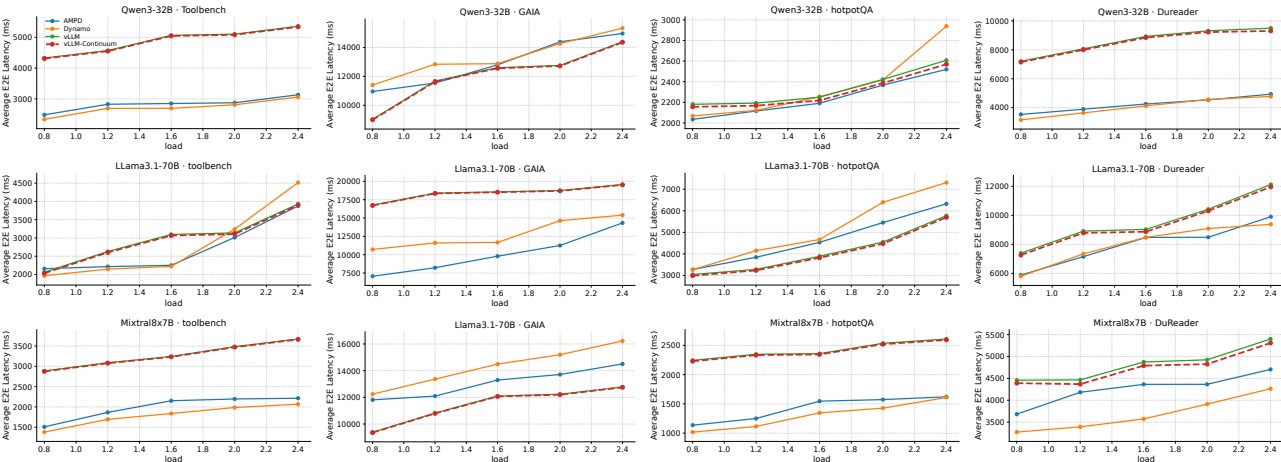

*Figure 10.* The average end-to-end (E2E) latency, corresponding to the experiments in Figure 5.

**Experimental results under stricter SLO requirements.** Figure 11 presents the SLO attainment under stricter TTFT and ITL requirements than those in Figure 5. It can be seen that AMPD remains consistently strong across all loads and thresholds, showing that AMPD's gains remain stable even under much stricter SLOs.

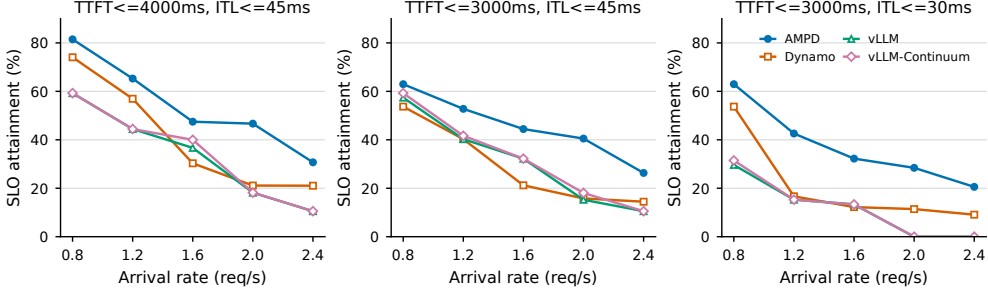

*Figure 11.* SLO attainment under different TTFT/ITL thresholds (Llama3.1-70B, DuReader). AMPD remains consistently strong across arrival rates and stricter SLO targets.

---

[7] https://github.com/OpenBMB/ToolBench.git
[8] https://huggingface.co/datasets/PatronusAI/TRAIL

