# OpenReview forum: "Efficient Multi-round LLM Inference over Disaggregated Serving"
_ICML.cc/2026/Conference — ICML 2026 regular_

### Official Review · Reviewer_Cqxk · 2026-02-17

**Soundness:** 4
**Presentation:** 3
**Significance:** 3
**Originality:** 4
**Overall Recommendation:** 4
**Confidence:** 4

**Summary:**

This paper presents AMPD for optimizing multi-round workloads in a disaggregated serving setting. It adaptively routes incremental prefills to either prefill or decode instances to balance KV cache transfer overhead against prefill–decode (PD) interference.

**Compliance With Llm Reviewing Policy:**

Affirmed.

**Final Justification:**

The paper studies routing optimization for multi-turn agents under PD separation, which is a timely and relevant problem. The approach appears sound and practically meaningful.

The rebuttal addressed my main concerns with quantitative results and improved my confidence in the work. However, I still find Section 5 somewhat disconnected from the rest of the paper, which affects overall clarity and cohesion.

Balancing the strengths and remaining presentation issue, I maintain a weak accept recommendation.

**Key Questions For Authors:**

1. How large is the KV cache transfer overhead relative to incremental prefill computation? Please report both the transferred KV size and the corresponding incremental prefill length.
2. Could you share any insights or references when choosing TTFT and ITL SLOs in the experiments? How sensitive are the results to different SLO configurations?
3. Did you consider prefix caching for prefill workers in your design?
4. Could you clarify the motivation and relevance of Section 5?

**Limitations:**

No. Authors could discuss the SLO configurations that the optimizations can benefit.

**Strengths And Weaknesses:**

Strengths
1. The KV cache transfer overhead in multi-round inference under PD disaggregation is an important and practical problem.
2. The proposed solution is simple yet practical.
3. The overall presentation is clear.

Weaknesses
1. Insufficient data support for the adaptive routing motivation. The paper mentions that existing PD-disaggregated systems always route non-initial prefills to prefill instances, leading to KV cache transfer overhead. However, no quantitative breakdown is provided to demonstrate the magnitude or impact of this overhead.
2. Lack of comparison with chunked prefill approaches (e.g., Sarathi-Serve [1]). PD disaggregation eliminates PD interference and enables stricter ITL SLOs, while AMPD reduces TTFT at the cost of slightly increased ITL. In scenarios with strict TTFT but relaxed ITL requirements, hybrid batching with chunked prefills may be a competitive or even superior solution. The paper neither discusses nor evaluates against this line of work.
3. The proposed routing algorithm ignores prefix caching and hit-rate. Prefill workers may have different cache hit rates, which can significantly affect prefill latency. A robust routing policy should explicitly consider cache locality and load balancing, as explored in systems such as Mooncake [2]. The paper does not discuss or compare against existing routing strategies in PD-disaggregated settings.
4. Unclear motivation and missing evaluation for Section 5. The rationale for deploying workers with heterogeneous TP sizes is not well explained. Concrete motivating scenarios illustrating the benefit would help. Moreover, this deployment planning appears largely disconnected from the multi-round workload optimization discussed earlier, and no experimental evaluation is provided.

References

[1] Agrawal A, Kedia N, Panwar A, et al. Taming the throughput–latency tradeoff in LLM inference with Sarathi-Serve. OSDI 2024.

[2] Qin R, Li Z, He W, et al. Mooncake: Trading more storage for less computation — a KVCache-centric architecture for serving LLM chatbots. FAST 2025.

---

> ### Author Rebuttal · Authors · 2026-03-31
>
> # W1 & Q1
>
> Thank you for the insightful comment. To quantify KV cache transfer overhead, we profile all incremental prefills in DuReader and GAIA with Llama-3.1-70B, measuring history length (l_hist), incremental input length (l_incr), KV transfer time (T_kv), and incremental prefill time (T_pre). We group tasks by T_kv/T_pre$ and summarize their proportions in the dataset by both task count and token count:
>
> ||T_kv/T_pre|Prop. (\#tasks)|Prop. (\#tokens)|Avg. l_hist|Avg. l_incr|Avg. T_kv|Avg. T_incr|
> |---|---|---|---|---|---|---|---|
> |DuReader|0%-1%|63.33%|52.00%|834.19|2477.04|3.79|730.22|
> ||1%–5%|17.78%|25.23%|4323.81|1399.50|6.56|450.31|
> ||5%–10%|7.78%|5.30%|2719.71|30.57|3.15|43.02|
> ||10%–20%|11.11%|17.47%|6317.90|24.00|7.27|51.57|
> |GAIA|0%–1%|60.34%|33.43%|828.66|956.51|2.05|307.02|
> ||1%–5%|25.86%|32.83%|3354.53|736.87|4.69|248.55|
> ||5%–10%|6.90%|14.41%|6541.50|191.25|7.71|103.84|
> ||10%–20%|5.17%|13.06%|8066.33|72.67|9.33|71.40|
> ||20%-25%|1.72%|6.26%|11691.00|18.00|13.42|65.23|
>
> When l_hist is large and l_incr is short (which is common in later rounds of multi-round workflows), KV transfer becomes non-negligible.
>
> Moreover, reducing KV transfer is only one motivation. Routing all prefills to prefill workers also increases the burden of prefill workers (as discussed in Sec. 4.1, lines 200–202). Our adaptive routing prevents prefill workers from being overwhelmed: in Fig. 5 (Right), it keeps 13.9%–31.7% of prefill tasks local on decode workers, reducing saturation and transfer overhead and improving overall SLO attainment by 27.37%–350%.
>
> ---
>
> # W2
>
> We appreciate this suggestion. We view chunked prefill as orthogonal and complementary to AMPD, not a competing alternative. Our work focuses on scheduling and deployment under multi-round interleaved workloads; chunked prefill instead slices a single prefill to reduce decode blocking. Since AMPD supports local prefill execution on decode workers, chunked prefill can be incorporated into AMPD to secure ITL as well. We will add this discussion in the revision.
>
> ---
>
> # Q2
>
> Since models/datasets differ substantially, it is common in LLM serving to set SLO requirements accordingly. Prior work also uses task-specific, empirically chosen SLOs rather than one fixed threshold: DistServe [OSDI'24] empirically sets TTFT targets from 0.125s to 15s across workload types (see Section 6.1 of DistServe's paper for more details); Llumnix [OSDI'24] and EPD [ICML'25] likewise use different latency targets.
>
> In our manuscript, we set thresholds to be moderate so that SLO attainment would not be near 0% or near 100%. We also evaluated more thresholds (reported in our response to Reviewer 9FVt due to space limits), showing that AMPD remains consistently strong across all thresholds, rather than only near the chosen point in our manuscript.
>
> ---
>
> # W3 & Q3
>
> We wish to clarify that the prefix cache is maintained on decode workers, not prefill workers, as noted in Footnote 4. (This follows Dynamo's design.) When a prefill task is routed as remote execution, the routed prefill worker reads the history KV cache from the decode worker, performs incremental prefill rather than full prefill, and sends back only the incremental KV cache. Thus, prefix caching is already considered in our design.
>
> Our method is also agnostic to cache management policy. If a request’s KV cache has been evicted, AMPD simply treats the incremental prefill as an initial prefill. Therefore, AMPD is compatible with arbitrary prefix-cache management schemes.
>
> Finally, storing the prefix cache on prefill workers instead of decode workers does not reduce total KV transfer in theory: although it may reduce decode-to-prefill transfer, the full KV cache must be repeatedly sent from the prefill worker to the decode worker after each incremental prefill.
>
> ---
>
> # W4 & Q4
>
> **Heterogeneous TP sizes.** Our formulation considers heterogeneous TP sizes to allow for flexible, irregular GPU allocation across phases. For example, in a 16-GPU cluster, the solver may explore three 2-GPU prefill workers. If TP sizes must be homogeneous, it would be forced to use five 2-GPU decode workers or ten 1-GPU decode workers. This rules out more flexible allocations such as two 4-GPU decode workers plus one 2-GPU decode worker, which can preserve more memory for decoding.
>
> **Relevance and evaluation of Sec. 5.** Sec. 5 is tightly coupled with Sec. 4. As noted in Footnote 5, the planner uses a simulator to predict P95 latency of candidate deployments. Appendix A.1 describes that the simulator follows the adaptive routing and prefill reordering from Sec. 4, so it aligns with the actual system runtime. We further validate both the efficiency and effectiveness of the planner in Appendix A.2 and A.3: it completes planning for 256 GPUs in about 1 minute, and the resulting deployments closely match the empirically optimal performance under different settings.
>
> We will add discussions in the revised manuscript.

---

> > ### Author Rebuttal · Reviewer_Cqxk · 2026-04-03
> >
> > Thank you for your response. For W2, I mean, could we directly use hybrid batching with chunked prefills to colocate PD instead of PD disaggregation?

---

> > > ### Author Response · Authors · 2026-04-03
> > >
> > > We thank Reviewer Cqxk for the insightful question. We agree that hybrid batching with chunked prefills can be powerful in regimes with strict TTFT requirements but relatively relaxed ITL requirements. However, it does not provide the same benefits as PD disaggregation in regimes with strict ITL requirements.
> > >
> > > To be specific, chunked prefill breaks a long prefill into smaller chunks, but prefill and decode still share the same worker resources and scheduler. As a result, the interference is mitigated rather than fundamentally removed. Particularly, under the multi-round LLM workloads we study, the interleaved prefill-decode pattern aggravates the interference, still degrading decode efficiency. In contrast, PD disaggregation separates the two phases to resolve the interference, and therefore secures ITL. Consequently, we focus on PD disaggregation in this work.
> > >
> > > Currently, PD disaggregation has become a widely used technique in LLM serving. Thus, we believe optimizing multi-round workflows under PD disaggregation is of significance. Besides, as discussed in our initial rebuttal, we view chunked prefill as complementary to our approach rather than a replacement.

---

### Official Review · Reviewer_suQR · 2026-02-27

**Soundness:** 3
**Presentation:** 3
**Significance:** 3
**Originality:** 3
**Overall Recommendation:** 4
**Confidence:** 2

**Summary:**

This paper studies multi-round LLM inference under prefill–decode (PD) disaggregated serving. In multi-round workflows (e.g., agents and iterative RAG), decoding is repeatedly interleaved with *incremental prefill* computations, creating an interleaved prefill–decode workload pattern that complicates scheduling and deployment under PD disaggregation. The authors propose AMPD, a serving framework that (1) plans prefill/decode resource allocation and parallelism offline via profiling and an ILP-based deployment planner, and (2) performs online adaptive scheduling via adaptive routing (decide local prefill on decode vs remote prefill on prefill workers) and TTFT-aware prefill queue reordering to improve SLO attainment under TTFT/ITL constraints. Experiments across multiple models and multi-round traces show improved SLO attainment relative to colocated and disaggregated baselines.

**Compliance With Llm Reviewing Policy:**

Affirmed.

**Final Justification:**

My concerns were addressed. I will raise my score to 4.

**Key Questions For Authors:**

1.How consistent is the offline planner’s surrogate objective with maximizing SLO attainment across traces/arrival rates and different TTFT/ITL thresholds? Are there regimes where it fails?

2.What is the simulator’s prediction error (P95 latency) relative to real measurements for representative configurations, and how robust is the final deployment choice to prediction noise?

3.What is the end-to-end overhead of online routing and queue reordering, and how does it scale with cluster size and concurrency?

**Limitations:**

yes

**Strengths And Weaknesses:**

Strengths：

1.Clear motivation aligned with real multi-round serving needs. The paper clearly identifies incremental-prefill interleaving as the core challenge when combining multi-round workflows with PD disaggregation, and frames it as a practical systems problem.

2.Online mechanisms are end-to-end and explicitly SLO-oriented. The adaptive routing mechanism explicitly trades off KV transmission/remote congestion versus decode interference, using windowed TTFT/ITL signals and a cost model to make actionable decisions; the prefill-side reordering uses a small lookahead window to reduce TTFT tail while preventing starvation.

3.Broad experiments and useful ablations. Evaluation spans multiple models and multi-round traces with both colocated and disaggregated baselines, and includes ablations isolating the effects of routing and queue reordering.



Weaknesses

1.The offline planner formulates an ILP to minimize the bottleneck worker’s worst-case P95 latency, while the system-level goal is to maximize SLO attainment (meeting both TTFT and ITL thresholds). The authors acknowledge this discrepancy, but the paper lacks a clear empirical or theoretical justification that minimizing this surrogate objective reliably improves SLO attainment under multi-round interleaved workloads, or when/why it could deviate from the SLO-optimal deployment.

2.The ILP parameters come from a simulator that models phase splitting, KV transmission, and even incorporates the incremental-prefill offload/local mechanism. However, the paper provides little quantification of simulator error magnitude, error sources, or how sensitive the final deployment choice and SLO attainment are to these errors.

3.Online routing reads globally shared windowed TTFT/ITL statistics and queue metadata (implemented via distributed shared memory/Redis). The paper claims the overhead is minor, but does not quantify routing overhead, nor how it scales with more workers or higher concurrency.

---

> ### Author Rebuttal · Authors · 2026-03-31
>
> # W1&Q1
>
> **Consistency of offline planner's results.** Indeed, there is a surrogate gap between the offline planner's objective (minimizing the bottleneck worker's P95 latency) and our system's goal (maximizing SLO attainment). Our design rationale is that under multi-round interleaved workloads, the tail latency of the bottleneck worker often determines whether it will trigger TTFT/ITL violations. Therefore, using it as a surrogate objective can effectively guide the deployment search in practice.
>
> In Appendix A.3 (Table 2), we empirically verified the effectiveness of the offline planner. The results show that under different settings, the top-3 ranking of deployment configurations given by the offline planner is consistent with the ranking measured by the real system. This indicates that although the surrogate objective is not perfectly equivalent to SLO attainment, it can reliably preserve the relative performance rankings of configurations, thereby effectively guiding deployment choices. We believe that such relative superiority among configurations is essential for systems work (which is also recognized by Reviewer 9FVt).
>
> **Regimes where it fails.** When the TTFT and ITL thresholds are extremely strict or extremely loose, the discrepancy in objectives may cause the offline planner's solution to deviate from the actual optimum. In these cases, the binary nature of the SLO metric dominates: the vast majority of deployment strategies will yield an SLO attainment of nearly 0% or 100%. Consequently, optimizing the continuous P95 latency metric will fail to translate into meaningful improvement in SLO attainment. However, in practical LLM serving scenarios, TTFT and ITL thresholds are usually set to reasonable targets, so our method is empirically reliable.
>
> ---
>
> # W2&Q2
>
> To address the reviewer's concern, we provide the actual P95 latency and the predicted P95 latency under different deployment configurations (Qwen3-32B, 1.6 reqs/s) below to demonstrate the prediction error.
>
> |Dataset|\# GPUs|Deployment Configurations|Actual P95 Latency (ms)|Predicted P95 Latency (ms)|
> |---|---:|---|---:|---:|
> |HotpotQA|8|P: TP=4,DP=1; D: TP=4,DP=1|7360|9855|
> |||P: TP=2,DP=2; D: TP=4,DP=1|7763|10151|
> |||P: TP=2,DP=1; D: TP=2,DP=3|7852|10405|
> |ToolBench|8|P: TP=4,DP=1; D: TP=4,DP=1|12540|15025|
> ||8|P: TP=2,DP=2; D: TP=4,DP=1|12991|15625|
> ||8|P: TP=2,DP=1; D: TP=2,DP=3|13228|16037|
> |DuReader|16|P: TP=4, DP=2;D: TP=4, DP=2|16020|19958|
> ||16|P: TP=4,DP=2; D: TP=2,DP=4|17535|21378|
> ||16|P: TP=2,DP=4; D: TP=4,DP=2|23362|29676|
> |GAIA|32|P:TP=8, DP=2; D: TP=8,DP=2|54878|63447|
> ||32|P: TP=8,DP=3; D: TP=8,DP=1|60275|67764|
> ||32|P: TP=4,DP=4; D: TP=8,DP=2|76420|80253|
>
> Following our response to W1&Q1 above, we wish to highlight again that since the primary goal of the offline planner is to identify top-performing deployment configurations from the candidates, it is more critical for the offline planner to be capable of preserving the relative ranking of deployment candidates, rather than minimizing the prediction errors. The results in Appendix A.3 (Table 2) of our manuscript show that the top-3 deployment configurations deduced by our planner are identical to the top-3 configurations evaluated in the real system. This empirically demonstrates that the prediction error does not harm the system performance.
>
> ---
>
> # W3&Q3
>
> The reviewer's comments are invaluable. We first empirically assess the overhead of adaptive routing (Section 4.1) and prefill reordering (Section 4.2), and then analyze their scalability.
>
> **Overhead.** We measured the overhead of our techniques:
> - The total time cost of adaptive routing is 0.39ms on average, with metadata reading (via distributed shared memory) consuming merely 0.15ms.
> - The total time cost of prefill reordering is 0.45ms on average.
>
> In contrast, LLM prefill computations and decoding steps typically take tens to hundreds of milliseconds. As a result, the end-to-end overhead of adaptive routing and prefill reordering is negligible. Moreover, they run on CPU and overlap with GPU computation.
>
> **Scalability.** When there are more GPUs, it is common to proportionally deploy more replicas (i.e., more workers) to increase concurrency.
> - For prefill reordering, since it is performed locally, its complexity is irrelevant to the number of GPUs.
> - Since the adaptive routing needs to iterate over the prefill workers (line 1 and line 7 of Algorithm 1), its complexity can be regarded as linear w.r.t. the number of GPUs. Nevertheless, given the extremely low overhead provided above, we believe it would not be a system bottleneck even for hundreds of GPUs. Moreover, at larger cluster scales, we can partition the GPUs into groups according to their locality, and perform the adaptive routing within each group, so that the overhead can be bounded.

---

> > ### Author Rebuttal · Reviewer_suQR · 2026-04-01
> >
> > Thank you for the response. This addresses my concerns, and I will raise my score accordingly.

---

> > > ### Author Response · Authors · 2026-04-01
> > >
> > > Thanks for your acknowledgement. We will incorporate the rebuttal in our revised manuscript.

---

### Official Review · Reviewer_YLd1 · 2026-03-10

**Soundness:** 3
**Presentation:** 3
**Significance:** 2
**Originality:** 2
**Overall Recommendation:** 4
**Confidence:** 4

**Summary:**

This paper targets the challenge of serving multi-round LLM inference under PD dis-aggregation. While PD dis-aggregation is effective for single-round inference, the authors observe that existing systems fail to properly handle the interleaved prefill–decode pattern induced by multi-round workflows. AMPD is proposed, which adopts a two-stage design: 1) an offline planning phase, which formulates a resource-constrained optimization problem (solved via LIP) to determine model placement and parallelization strategies; and 2) an online adaptive scheduling phase, which dynamically routes incremental prefill tasks either to the local worker or to remote ones.

**Compliance With Llm Reviewing Policy:**

Affirmed.

**Final Justification:**

Final recommendation upon rebuttal: weak accept, improved score

Strengths:
The paper addresses a practical problem: efficiently serving multi-round LLM under PD dis-aggregation is highly relevant for agentic and retrieval-augmented applications. The authors identify and articulate the overlooked interleaved prefill–decode pattern. AMPD is proposed via offline planner and online router.

Weaknesses:
1. The authors have to add more discussion on the workflow, regarding whether the history KV is used and transferred from decode to prefill worker, also its evaluations.
2. The re-planning requires extra communication and overhead, which should be also discussed (e.g., whether maintaining the connections has the limit). More related works have to be studied and cited.
3. The online router depends on windowed TTFT and some statistics, but current description remains intuitive. The following should be well studied: How is the window size determined, and what is its impact on system stability? How is the ``sufficient slack'' windowed ITL formally defined?

**Key Questions For Authors:**

1. The authors should illustrate more implementation details on ``the prefill worker reads the history KV cache from the responsible decode worker if necessary''.
2. Is it suitable to directly offload all prefill tasks to remote workers? The analysis (e.g., $\alpha$ and $\beta$ in algorithm 1) is missing for various strategies (local, remote or the interleaved impact), especially under the consideration of prompt length (input), decoding length (output), bandwidth, etc.
3. Some notations should be revised. For example, {$local, remote_1, remote_2, ...$} can be replaced with a set notation.
4. Some other works like jointly optimizing aggregation and dis-aggregation should be also considered (or as the baseline for analysis).

**Limitations:**

yes

**Strengths And Weaknesses:**

Strengths:
The paper addresses a practical problem: efficiently serving multi-round LLM under PD dis-aggregation is highly relevant for agentic and retrieval-augmented applications. The authors identify and articulate the overlooked interleaved prefill–decode pattern. AMPD is proposed via offline planner and online router.

Weaknesses:
1. The author should discuss more about the remote execution. For example, the details about ``the prefill worker reads the history KV cache from the responsible decode worker if necessary'' should be studied. How to implement it? And, whether it is possible to pre-fetch the KV cache in advance, since the offline planner has already considered the status.
2. The paper should also study more about the interleaved impact. For example, is it suitable to offload all prefill tasks to remote workers, to directly avoid the interleaved impact (i.e., decode workers only focusing on decoding)? The analysis is missing under the consideration of prompt length, decoding tokens, bandwidth, etc.
3. The offline deployment is formulated as a resource-constrained integer linear programming (ITL). However, it lacks the analysis on the scale and computational complexity (especially under realistic cluster size).
4. The online router depends on windowed TTFT and some statistics, but current description remains intuitive. How is the window size determined, and what is its impact on system stability? How is the ``sufficient slack'' windowed ITL formally defined?

---

> ### Author Rebuttal · Authors · 2026-03-31
>
> # W1&Q1
>
> We thank the reviewer and clarify details of remote execution and KV cache transfer.
>
> **Clarification of "if necessary".** As introduced in Sec. 3 (lines 132-138), each request (a.k.a. session) is first bound to a decode worker, which manages its KV cache and all decode steps. For the initial prefill task, if it is executed remotely, no history KV cache needs to be read. For subsequent incremental prefill tasks, if remote execution is chosen, the prefill worker must read the history KV cache from the decode worker. This is what "if necessary" means.
>
> **Implementation of KV cache reads.** As described in Sec. 6, we use NVIDIA NIXL for P2P RDMA communication and implement KV transmission as lazy reads. When adaptive routing chooses remote execution for an incremental prefill task, the decode worker first sends only metadata to the prefill worker, including the request id, history length, and NIXL memory handle. The prefill worker does not fetch KV immediately; it reads the KV cache only when queue reordering schedules this incremental prefill as the next task.
>
> **Prefetching is already implemented.** As described in Sec. 6, KV transmission overlaps with previous-task computation. While GPU is executing the current prefill, the queue reordering runs on a background CPU thread, deciding the next task and triggering KV reading. Therefore, KV transfer is effectively prefetched.
>
> ---
>
> # W2&Q2
>
> **Why not offload all prefill tasks remotely?** Always routing prefill tasks to remote prefill workers falls back to static PD disaggregation, which AMPD is designed to avoid. This can improve ITL because decode workers focus on steady-state decoding, but it also incurs extra KV-transfer cost and increases prefill-worker load, which harms TTFT. Hence, the system should adaptively choose local vs. remote execution based on real-time load, rather than using a fixed strategy. This is a core design goal of AMPD.
>
> In end-to-end experiments, AMPD substantially outperforms Dynamo, which always routes prefill tasks remotely, and the ablation study in Figure 5 further validates the benefit of adaptive routing.
>
> **Length information and bandwidth are considered.** As described in Sec. 3 (lines 154-164), we use a piecewise alpha-beta model to build time cost functions:
> - $T_{pre}(l_{hist}, l_{incr}; \theta)$ models prefill time as a function of history length and incremental input length.
> - $T_{kv}(l_{ctx}; \theta_{src}, \theta_{dst})$ models KV cache transfer time given length and parallel strategy, which also considers bandwidth.
>
> In Sec. 4, Alg. 1 and Eq.(1)–(2) use these functions to estimate the cost of local vs. remote execution, so length information and bandwidth are already considered.
>
> ---
>
> # W3
>
> The deployment problem (Eq.(5)) is an ILP, so it is NP-hard in general. However, our search space is constrained: the decision variables only represent the numbers of worker replicas under a small set of model parallel degrees. In practice, existing solvers (e.g., SCIP library) can solve it efficiently.
>
> As reported in Appendix A.2 (Figure 7), the offline planning time ranges from 0.54s to 61.67s when scaling from 8 to 256 GPUs. Thus, even at 256 GPUs, planning takes only about 1 minute, which is negligible for online serving. This can be further accelerated via multi-threading, which is supported by solvers like SCIP.
>
> ---
>
> # W4
>
> We will clarify the online router and windowed statistics.
>
> As described in Sec. 3 (lines 124-125), each prefill/decode worker records its windowed TTFT/ITL statistic, by default the average over the past 10 seconds. Recording these statistics itself does not affect system stability. (Besides, the window size $w$ is used in prefill reordering (Sec. 4.2), rather than in recording windowed statistics.)
>
> As described in Sec. 4.1 (lines 2–5 of Alg. 1), we formally identify "sufficient slack" by: $\widehat{TTFT}_i \le \alpha \cdot TTFT_{thres}$ and $\widehat{ITL} \le \beta \cdot ITL_{thres}$, where
> - $\widehat{TTFT}_i$ is the windowed TTFT of prefill worker $i$,
> - $\widehat{ITL}$ is the windowed ITL of the current decode worker,
> - $TTFT_{thres}$ and $ITL_{thres}$ are the SLO thresholds,
> - $\alpha,\beta$ are hyper-parameters used to determine TTFT/ITL slack.
>
> ---
>
> # Q3
>
> Thank you for the suggestion. We will revise the notation for better readability.
>
> ---
>
> # Q4
>
> Thank you for the suggestion. We will expand discussion of related systems such as DynaServe (arXiv:2504.09285) and TaiChi (arXiv:2508.01989), which jointly optimize aggregation and disaggregation. However, these works do not consider the interleaved prefill-decode pattern in multi-round workflows. For instance, they assign part of the decode workload to prefill-related instances, which is less suitable here because prefill workloads are much heavier in multi-round workflows compared to single-round settings. Besides, these systems are not open-sourced, so direct comparison is non-trivial. We will add discussion in the revision.

---

> > ### Author Rebuttal · Reviewer_YLd1 · 2026-04-02
> >
> > Thank you for your response. My concerns have been partially addressed. I will maintain my score at 3.
> >
> > ``The offline planning time ranges from 0.54s to 61.67s.'' Whether all possible conditions are resolved within such one minute? I think the dynamics of the system (batched requests, KV occupation, etc.) have the influence on the decisions. And, planning using one minute is too long for online serving (e.g., 2 seconds SLO for TTFT).

---

> > > ### Author Response · Authors · 2026-04-03
> > >
> > > We thank Reviewer YLd1 for the reply. Below we respond to the two follow-up questions/concerns individually.
> > >
> > > ---
> > >
> > > Firstly, to address Reviewer YLd1's question about the offline planning time under different conditions, we provide the offline planning time for Llama3.1-70B (the largest evaluated model) on more datasets and under different request arrival rates. (Note that the differences in sequence lengths across datasets (see Table 1 of our manuscript) and request arrival rates affect the conditions of batching and KV cache occupation.) It can be seen that the offline planning time is relatively stable when the number of GPUs is fixed.
> > >
> > > | Model | Trace | Arrival Rate (reqs/s) | 8 GPUs | 16 GPUs | 32 GPUs | 64 GPUs | 128 GPUs | 256 GPUs |
> > > |---|---|---|---:|---:|---:|---:|---:|---:|
> > > | Llama3.1-70B | DuReader | 0.8 | 0.53 | 1.92 | 4.99 |14.40  |28.53  | 59.90 |
> > > | Llama3.1-70B | DuReader | 1.2 | 0.52 | 1.91 |5.04  |13.88  | 28.64  |60.13  |
> > > | Llama3.1-70B | DuReader | 1.6 | 0.47 | 1.89 | 5.06 | 14.34 | 28.06 |60.62  |
> > > | Llama3.1-70B | DuReader | 2.0 |0.52 | 1.93 | 5.02 | 14.50 | 27.95 | 61.43 |
> > > | Llama3.1-70B | DuReader | 2.4 | 0.48 |  1.90|  5.05 | 14.31 | 27.85 |60.73  |
> > > | Llama3.1-70B | HotpotQA | 0.8 | 0.51 | 1.88 | 4.99 | 14.35 | 28.88 | 61.56 |
> > > | Llama3.1-70B | HotpotQA | 1.2 | 0.53 |  1.86| 5.02 | 14.34 | 28.58 |61.65  |
> > > | Llama3.1-70B | HotpotQA | 1.6 | 0.52 | 1.93 | 5.06 | 13.92 | 28.74 |60.94  |
> > > | Llama3.1-70B | HotpotQA | 2.0 | 0.54 | 1.91 | 4.97 |14.52  | 28.92 |60.88  |
> > > | Llama3.1-70B | HotpotQA | 2.4 | 0.52 | 1.94 |4.99  | 14.30 | 28.61 | 61.75 |
> > > | Llama3.1-70B | GAIA | 0.8 | 0.47 | 1.94 | 4.97 | 14.38 |28.07  | 59.79 |
> > > | Llama3.1-70B | GAIA | 1.2 | 0.54 | 1.91 | 5.03 | 14.26 | 28.84 | 61.26 |
> > > | Llama3.1-70B | GAIA | 1.6 | 0.53 | 1.89 | 5.10 |14.44  |27.62  | 60.54 |
> > > | Llama3.1-70B | GAIA | 2.0 | 0.52 | 1.96 | 4.91 | 14.34 | 28.54 | 61.42 |
> > > | Llama3.1-70B | GAIA | 2.4 | 0.54 | 1.93 | 5.06 | 14.50 | 28.88 | 61.67 |
> > > | Llama3.1-70B | ToolBench | 0.8 | 0.54 |1.92  |5.06  | 13.96 |28.74  | 61.22 |
> > > | Llama3.1-70B | ToolBench | 1.2 | 0.51 | 1.93 | 5.01 | 14.50 |28.86  |61.66  |
> > > | Llama3.1-70B | ToolBench | 1.6 | 0.48 | 1.90 | 5.08 | 14.26 | 29.10 | 60.34 |
> > > | Llama3.1-70B | ToolBench | 2.0 | 0.55 | 1.93 | 5.04 | 14.44 | 27.68 | 60.83 |
> > > | Llama3.1-70B | ToolBench | 2.4 | 0.51 | 1.89 | 5.02 | 14.36  | 27.94 | 61.36 |
> > >
> > >
> > >
> > >
> > > ---
> > >
> > > Secondly, regarding Reviewer YLd1's concern about the feasibility for online serving, we would like to clarify that the offline planning does not need to be triggered for each incoming request. Instead, it is triggered at initialization or when replanning is required. Specifically, when the workload pattern (e.g., input/output length, request arrival rate) changes substantially during online serving, the current deployment configuration may not be optimal, so we would need to trigger the offline planning to update the deployment configuration. This is analogous to prior online monitor-and-replan approaches such as DistServe (e.g., see Section 4.3 of DistServe's paper).
> > >
> > > However, during replanning, the online service is not halted, since the offline planning process runs on a background CPU thread. Moreover, once a new deployment configuration is obtained after the replanning process, we can perform rolling migration (adjusting replicas one by one) to avoid service interruption.
> > >
> > > We agree that dynamic replanning is indeed an important and broad topic, however, it is orthogonal to the main focus of our work. In the revision, we will add more discussion about replanning to address Reviewer YLd1's concern.

---

### Official Review · Reviewer_9FVt · 2026-03-13

**Soundness:** 3
**Presentation:** 3
**Significance:** 3
**Originality:** 3
**Overall Recommendation:** 4
**Confidence:** 3

**Summary:**

The authors tackle a real problem with serving LLMs for multi-round workflows (i.e., agents calling tools or RAG systems fetching documents mid-inference): existing disaggregated serving systems were designed for single-round inference and handle the interleaved prefill/decode pattern of multi-round inference poorly. AMPD addresses this with two main ideas:

- An online scheduling mechanism that adaptively decides whether incremental prefill tasks should run locally on the decode worker or be routed to a dedicated prefill worker, combined with a reordering policy to maximize SLO attainment
- An offline planner that uses integer linear programming to find the optimal GPU allocation and parallelism strategy for both worker types, accounting for multi-round workload characteristics upfront

Taken together these yield substantial SLO attainment improvements over both co-located and disaggregated baselines across a range of models and workloads

**Compliance With Llm Reviewing Policy:**

Affirmed.

**Final Justification:**

The responses addressed my concerns and I would like to keep my positive score.

**Key Questions For Authors:**

1. **SLO threshold selection:** The thresholds vary a lot across settings and it's not clear how they were chosen. How sensitive are the results to these choices? If AMPD's advantage is concentrated in a narrow band around the chosen thresholds, that would notably change our read of the results.

2. **Hyperparameter tuning:** Were α=0.9, β=0.85, w=3 selected before or after seeing the evaluation results? If these were tuned on the same workloads used for evaluation, that's a fairness concern worth addressing.

3. **vLLM-Continuum underperformance:** vLLM-Continuum was designed specifically for multi-round workflows, yet it barely beats vanilla vLLM in most settings. Why? Is this a fundamental limitation of co-located architectures, or something specific to these workloads? The answer would help clarify how much of AMPD's gains come from disaggregation itself vs. the scheduling on top.

4. **Static deployment assumption:** The offline planner assumes a fixed workload distribution. In practice workloads shift. how often would replanning be needed, and what happens to SLO attainment during the switchover?

5. **Scalability of KV transmission:** All experiments use at most 32 GPUs over 200GB/s InfiniBand. Does the approach still hold up at larger scales where KV transmission overhead could become a more significant bottleneck?

**Limitations:**

yes

**Strengths And Weaknesses:**

### Strengths

- **The problem identification is sharp and well-motivated.** The observation that PD disaggregation and multi-round inference have been studied independently but never properly integrated is a genuine gap, and the paper makes a convincing case that this gap causes real performance problems. The breakdown analysis in Figure 4 — showing how co-located and disaggregated baselines each get "trapped" by either ITL or TTFT constraints — is particularly effective at building intuition for why the problem is hard.

- **The two-part solution (adaptive routing + prefill reordering) is clean and well-matched to the problem structure.** The routing algorithm's fallback logic (check prefill slack, then decode slack, then estimate and compare costs) is sensible and the cost estimation formulas are clearly derived from first principles.

- **The offline planner is a nice contribution.** Framing resource allocation as an ILP and validating that the planner's top-ranked configurations match real-system serving (Table 2, with perfect agreement across all model/trace combinations) is genuinely reassuring and not something every systems paper bothers to verify this carefully.

- **Empirical coverage is broad.** Testing across three model families (dense, MoE, large dense), four workload traces with meaningfully different characteristics, and multiple request arrival rates gives reasonable confidence that the results are not cherry-picked.

- **The ablation study (Figure 5) is useful and honest.** Separating the contributions of adaptive routing and prefill reordering, and showing the distribution of local vs. remote execution decisions across workloads, gives the reader a real sense of where the gains are coming from.

### Weaknesses

- **The SLO thresholds feel somewhat arbitrary and are not well justified.** The thresholds vary widely across models and traces (e.g., TTFT of 130ms for Mixtral ToolBench vs. 5000ms for Llama DuReader), and it is not clear how sensitive the relative rankings of systems are to these choices. A reader might reasonably wonder whether AMPD's advantage shrinks or grows under tighter or looser thresholds.

- **The adaptive routing algorithm introduces three hyperparameters (α, β, and window size w), and the sensitivity analysis is limited.** While Figure 6 is appreciated, it only covers one model/trace/load combination. It is unclear whether the same moderate values work well across the full range of settings tested, or whether the reported defaults were tuned on the evaluation workloads.

- **The paper largely sidesteps fault tolerance and dynamic scaling.** In real deployments, workers can fail or be added, and workload characteristics shift over time. The offline planner assumes a static configuration, and it is not discussed how often replanning is needed or what happens during transitions. This is understandable as a scope limitation but worth acknowledging more explicitly.

- **The comparison to vLLM-Continuum deserves more discussion.** The paper notes that vLLM-Continuum often performs comparably to vanilla vLLM, which is a somewhat surprising finding given that it was specifically designed for multi-round workflows. A deeper analysis of why it underperforms expectations would strengthen the paper's positioning and be useful to the community.

- **Figure 4 is dense and hard to parse at a glance.** The bottom row breakdown (initial TTFT, incremental TTFT, ITL) is one of the most informative parts of the evaluation but is easy to miss or misread given the layout.

---

> ### Author Rebuttal · Authors · 2026-03-31
>
> # W1&Q1
> Thank you for raising this concern. Varying SLOs across models/datasets is standard in LLM serving because model sizes and input/output lengths differ substantially. Prior work also uses task-specific, empirically chosen SLOs rather than one fixed threshold: DistServe [OSDI'24] empirically sets TTFT targets from 0.125s to 15s across workload types (see Section 6.1 of DistServe's paper for more details); Llumnix [OSDI'24] and EPD [ICML'25] likewise use different latency targets.
>
> Below, we provide results under different thresholds (Llama3.1-70B, DuReader). AMPD remains consistently strong across all loads and thresholds, rather than only near the chosen point in our manuscript. Thus, AMPD’s gains remain stable even under much stricter SLOs, including regimes where all methods have low attainment.
>
> ||reqs/s|Dynamo|AMPD|vLLM|vLLM-Continuum|
> |---|---|---|---|---|---|
> |TTFT≤4000ms,ITL≤45ms|0.8|74.07%|81.48%|59.26%|59.26%|
> ||1.2|56.94%|65.28%|44.44%|44.44%|
> ||1.6|30.30%|47.47%|36.67%|40.00%|
> ||2.0|21.11%|46.67%|18.18%|18.18%|
> ||2.4|21.05%|30.70%|10.53%|10.53%|
> |TTFT≤3000ms,ITL≤45ms|0.8|53.70%|62.96%|57.41%|59.26%|
> ||1.2|40.28%|52.78%|40.28%|41.67%|
> ||1.6|21.21%|44.44%|32.22%|32.22%|
> ||2.0|15.79%|40.51%|15.25%|18.06%|
> ||2.4|14.44%|26.32%|10.53%|10.53%|
> |TTFT≤3000ms,ITL≤30ms|0.8|53.70%|62.96%|29.63%|31.48%|
> ||1.2|16.67%|42.59%|15.28%|15.28%|
> ||1.6|12.22%|32.24%|13.33%|13.33%|
> ||2.0|11.40%|28.43%|0.00%|0.00%|
> ||2.4|9.09%|20.57%|0.00%|0.00%|
>
> ---
>
> # W2&Q2
> We clarify that the default hyperparameters (α=0.9, β=0.85, w=3) were not tuned per evaluation workload. Instead, the same setting is used for all end-to-end experiments. (In fact, w=3 is not the best choice in Fig. 6.) We chose moderate values to better trade off hardware utilization and congestion detection: as discussed in the last paragraph of Section 7, smaller α detects prefill congestion earlier but may underutilize prefill workers, while smaller β detects decode congestion earlier but may reduce decode utilization.
>
> To address generality, we added sensitivity results on a different setting (Qwen3-32B, HotpotQA, 1.2 req/s), which show trends similar to Fig. 6:
> - w=2/3/4/5: attainment(%)=88.89/86.67/87.78/87.78
> - α=0.75/0.80/0.85/0.90/0.95/1.00: attainment(%)=67.78/71.11/81.11/86.67/77.78/73.33
> - β=0.75/0.80/0.85/0.90/1.00: attainment(%)=85.56/85.56/86.67/75.56/65.56
>
> These results show that the default hyper-parameters are robust, not workload-specific tuning.
>
> ---
>
> # W3&Q4
> We agree fault tolerance, elastic scaling, and workload drift are important. In the revision, we will state more clearly that our offline planner assumes a fixed workload distribution, while node failures, elastic scaling, and long-term drift are outside the main scope of this paper.
>
> In practice, replanning need not occur with a fixed frequency; it can be triggered by monitored statistics. Since AMPD already maintains windowed TTFT/ITL statistics for online routing, workload drift can be detected from these metrics and used to trigger replanning when necessary, similar to prior online monitor-and-replan approaches such as DistServe (see Section 4.3 of DistServe's paper for more details).
>
> During switchover, SLO attainment may temporarily decrease. To reduce impact, deployment changes can use rolling migration (adjusting replicas one by one), avoiding service interruption.
>
> ---
>
> # W4&Q3
> As discussed in Sec. 7.2 (lines 423-431), vLLM-Continuum uses a co-located architecture with session-aware queuing. By prioritizing incremental prefills, it improves KV reuse and can reduce TTFT. However, prefill and decode still share the same hardware, so compute-bound prefill continues to interfere with memory-bound decode.
>
> Therefore, although vLLM-Continuum often improves TTFT, its ITL remains close to vanilla vLLM, as seen in the bottom row of Fig. 4. In multi-round workflows, when ITL becomes the key bottleneck for meeting SLOs, TTFT-only improvements do not consistently translate into higher end-to-end SLO attainment.
>
> AMPD’s gains come from both disaggregation and scheduling: disaggregation protects decode workers from severe interference and improves ITL, while adaptive routing and prefill reordering improve TTFT that standard disaggregation may otherwise hurt.
>
> ---
>
> # Q5
> At larger scales, KV transmission is more costly if cross-node bandwidth is low. However, in typical data centers, a single high-bandwidth pod (connected via InfiniBand) often contains hundreds of GPUs, which is sufficient for many LLM deployments, so KV transfer would not be the dominant bottleneck in practice. Moreover, we can incorporate locality-aware deployment and routing: partition GPUs into groups based on locality, deploy within each group, and perform routing only within a group so KV does not cross slow long-haul links.
>
> ---
>
> # W5
> Thank you for pointing this out. We will revise Fig. 4 to make the bottom-row TTFT/ITL breakdown easier to read.

---

> > ### Author Rebuttal · Reviewer_9FVt · 2026-04-03
> >
> > Thank you. This addressed my questions.

---

> > > ### Author Response · Authors · 2026-04-03
> > >
> > > We thank Reviewer 9FVt for the acknowledgement. We will incorporate the rebuttal in our revised manuscript.

---

### Decision · Program_Chairs · 2026-04-30

**Decision:**

Accept (regular)

**Comment:**

This paper presents AMPD, a disaggregated serving framework designed specifically for multi-round Large Language Model (LLM) inference. Recognizing that existing prefill-decode (PD) disaggregation systems struggle with the interleaved prefill and decode patterns inherent in multi-round workflows like autonomous agents and RAG, the authors propose a two-stage solution. The framework combines an offline integer linear programming (ILP) planner to determine optimal GPU allocation and parallelism, with an online adaptive scheduling mechanism that dynamically routes incremental prefill tasks to either local decode workers or dedicated prefill workers to maximize Service Level Objective (SLO) attainment.

The reviewers unanimously agreed that the paper targets a highly practical, well-motivated, and largely overlooked problem in LLM serving (9FVt, YLd1, suQR, Cqxk). The two-part solution is praised for being clean, sensible, and effectively matched to the problem structure (9FVt, suQR). Furthermore, the reviewers appreciated the broad empirical coverage across various model families and workloads, noting that the combination of offline planning and online adaptive routing yields substantial improvements in SLO attainment over existing baselines (9FVt, YLd1, suQR). The inclusion of insightful ablation studies and the empirical validation of the offline planner were also highlighted as major strengths (9FVt, suQR).

During the review process, reviewers raised several valid concerns, primarily regarding the justification of the selected SLO thresholds and routing hyperparameters (9FVt, suQR, Cqxk), as well as requests for deeper implementation details on KV cache transmission overhead and remote execution (YLd1, Cqxk). Additionally, there were questions about the offline planner's scalability, its surrogate objective mismatch (YLd1, suQR), and the lack of comparisons against chunked prefill or prefix caching baselines (Cqxk). In the rebuttal, the authors satisfactorily addressed these points by providing additional quantitative breakdowns of the KV cache transfer overhead, clarifying the robustness of their hyperparameter choices across different scales, and offering extended discussions on how their system interacts with complementary techniques like chunked prefill, thereby alleviating the reviewers' primary concerns.

Ultimately, this paper makes a solid and timely contribution to the systems community by effectively bridging the gap between multi-round inference workflows and PD disaggregated serving. The identified problem is significant, the proposed methodologies are structurally sound, and the empirical results convincingly demonstrate the efficacy of the framework. Given the strong foundational ideas and the authors' adequate resolution of the evaluation and implementation queries during the rebuttal phase, the merits of this work clearly outweigh its limitations. Therefore, I recommend that this paper be accepted.